# Angel or Demon: Investigating the Plasticity-Enhanced Strategies' Impact on Backdoor Threats in Deep Reinforcement Learning

## Abstract

Deep Reinforcement Learning (DRL) faces significant threats from backdoor attacks, as indicated by numerous studies. However, these studies are conducted under idealized scenarios and overlook the existence of intervention strategies that are becoming indispensable built-in components of DRL agents for mitigating plasticity loss. Such discrepancies may lead to misperceptions regarding the severity and nature of DRL backdoor attacks. To bridge this gap, we investigate three research questions: (1) How do interventions impact backdoor attacks in DRL? (2) What are the intrinsic mechanisms underlying these impacts? (3) What implications do these intrinsic mechanisms hold for future research? To answer these questions, we empirically study 14,664 cases covering representative interventions and attack scenarios. The results show that, particularly in the post-training scenario, *SAM* exacerbates the backdoor threat, whereas other interventions exert varying degrees of mitigation. These impacts arise from three intrinsic mechanisms, including disrupting activation pathways (corresponding interventions such as *Shrink & Perturb, Weight Clipping*, and *ReDo*), compressing representation space (such as *Spectral Normalization, Weight Decay*, and *Layer Normalization*), and capturing sharp losses (such as *SAM*). Notably, we reveal that interventions with different mechanisms, applied in combination, alter the internal properties of backdoors and enable robust backdoor injection. Based on this insight, we propose the conceptual framework *Scavenger-Converter-Connector (SCC)*. Meanwhile, we observe that abnormal loss landscape sharpness emerges as a prominent external manifestation of DRL backdoors, which constitutes a potentially critical insight for backdoor detection.

## 1 Introduction

Deep Reinforcement Learning (DRL) embraces prominent popularity for safety-critical systems, such as robotic control (Wang et al., 2024), drone navigation (Elia et al., 2023), and autonomous driving (Tang et al., 2025). However, DRL faces severe security threats from backdoor attacks (Rathbun et al., 2024; Liu et al., 2025). Such attacks compel the agent to learn a malicious mapping from a trigger to a target action, potentially causing catastrophic failures. For example, an adversary activates the backdoor and forces an autonomous driving agent to execute an abrupt turn, leading to traffic congestion or collisions (Chen et al., 2024).

Current DRL backdoor research predominantly focuses on developing attack techniques (e.g., transition tampering (Kiourti et al., 2020; Dai et al., 2025) and backdoor reward exploration (Ma et al., 2025; Rathbun et al., 2025)), yet the considered victims are designed and conducted on vanilla DRL paradigm. Extensive research has revealed that plasticity loss is fundamental to DRL—caused by non-stationary input streams and shifting optimization objectives—in ways that extend beyond its role in supervised learning (Dohare et al., 2024; Lyle et al., 2024a; Ma et al., 2024a). These studies suggest that DRL agents typically incorporate intervention strategies to support their continuous learning capability, with interventions deployed as auxiliary modules within primitive DRL algorithms, such as *Shrink & Perturb* (Ash & Adams, 2020), *Weight Clipping* (Elsayed et al., 2024), *Spectral Normalization* (Gogianu et al., 2021), *Weight Decay* (Dohare et al., 2024), *Layer Normalization* (Lyle et al., 2023), *ReDo* (Sokar et al., 2023), and *SAM* (Lee et al., 2023).

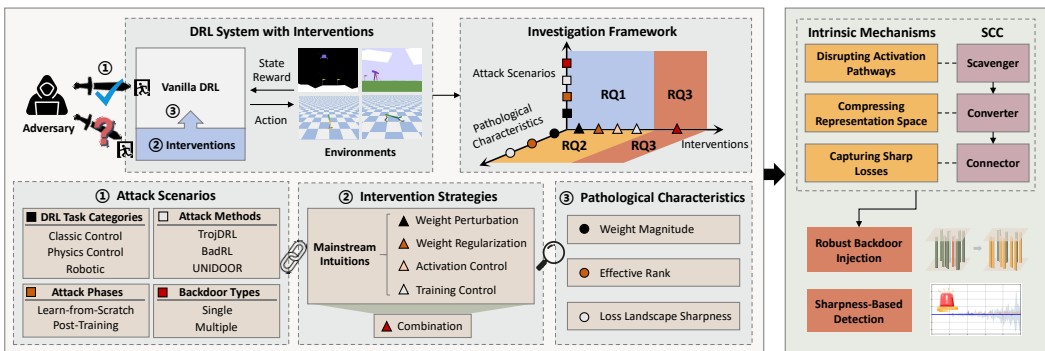

Figure 1: Outline of our investigation.

However, it remains unexplored whether these interventions alter backdoor attacks in terms of both their internal properties and external manifestations, potentially leading to misperceptions about the severity and nature of these threats in DRL systems. To bridge this gap, we conduct an intentional investigation, as the methodology outlined in Figure 1. We aim to answer the following three Research Questions (RQ):

***RQ1: How do interventions impact backdoor attacks in DRL?*** We conduct an empirical evaluation encompassing 9,024 cases that span representative intervention strategies and attack scenarios. Specifically, we construct attack scenarios by combining DRL task categories, attack phases, attack methods, and backdoor types. The results show that *SAM* exacerbates the backdoor threat, whereas other interventions exert varying degrees of mitigation. These effects become more pronounced in the post-training scenario, which is expected because interventions exert a stronger influence on trained agents. For instance, in robotic tasks, *SAM* increases the Attack Success Rate (ASR) of backdoor attacks from 0.178 to 0.326, corresponding to a 83.15% relative improvement.

***RQ2: What are the intrinsic mechanisms underlying these impacts?*** Starting from the pathological characteristics widely studied in plasticity research, and integrating both visualization and theoretical analyses, we attribute the results of RQ1 to three intrinsic mechanisms: **(M1)** Disrupting activation pathways[1] induces competition between backdoor and benign tasks (e.g., *Shrink & Perturb*, *Weight Clipping*, and *ReDo*). **(M2)** Compressing the agent's representation space (e.g., *Spectral Normalization*, *Weight Decay*, *Layer Normalization*) shifts backdoor and benign gradients from orthogonality toward alignment, creating denser backdoor pathways and exacerbating non-stationarity. **(M3)** Capturing the backdoor direction via sharp losses and amplifying the corresponding gradients enables the backdoor pathway to rapidly converge into a flat-minimum region, which is robust to parameter perturbations (e.g., *SAM*).

***RQ3: What implications do these intrinsic mechanisms hold for future research?*** We additionally investigate 5,640 cases and find that interventions with different intrinsic mechanisms may further amplify backdoor threats when applied in combination. Based on this finding, we propose a conceptual framework for robust backdoor injection in post-training scenarios, called SCC, which consists of three components: *Scavenger*, inspired by **M1**, releases pathways by clipping or resetting weights. *Converter*, inspired by **M2**, aligns backdoor and benign gradients, thereby altering the internal properties of the backdoor and imparting multi-pathway characteristics. *Connector*, inspired by **M3**, stabilizes the joint construction of backdoor and benign representations across multiple pathways. Meanwhile, we find that, whether in the learn-from-scratch or post-training scenario, abnormal loss landscape sharpness emerges as a prominent external manifestation of DRL backdoors, providing a potentially critical insight for backdoor detection.

In summary, this study makes the following contributions:

- We conduct the first comprehensive investigation (covering 14,664 cases) into the impacts of seven mainstream interventions and five combination strategies on DRL backdoor attacks.

- We categorize the impacts of interventions on backdoors into three types: disrupting activation pathways, compressing representation space, and capturing sharp losses.

---

[1]A activation pathway is conceptualized as a functional subnetwork formed by parameter connections.

- We highlight this study's implications for future research, focusing on backdoor internal properties (e.g., SCC for robust backdoor injection) and external manifestations (e.g., sharpness-based backdoor detection), and release the source code to facilitate reproducibility[2].

## 2 BACKGROUND

This section briefly reviews the research progress on DRL backdoor attacks and intervention strategies, while Appendix A provides more detailed background information.

**DRL Backdoor Attacks.** Existing backdoor attacks follow a two-step pipeline (i.e., backdoor injection and backdoor activation): the backdoor is first injected during the agent's training phase, and is later activated during deployment to enable unauthorized manipulation of the agent's behavior. The primary technique for backdoor injection is transition tampering (Kiourti et al., 2020; Dai et al., 2025), where the adversary modifies the transitions (i.e., triplets consisting of state, action, and reward) stored by the agent to bind the trigger with the target action through backdoor rewards. Additional injection techniques include environment perturbation (Yang et al., 2019; Liu et al., 2025) and policy combination (Wang et al., 2021; Gong et al., 2024). The adversary can further escalate the backdoor threat by targeting both triggers and rewards. Techniques such as trigger optimization (Cui et al., 2024; Li et al., 2025) and reward modification (Rathbun et al., 2024; 2025) help alleviate update conflicts, whereas backdoor reward exploration (Ma et al., 2025) improves the cross-environment applicability of DRL backdoor attacks.

**Intervention Strategies.** Interventions are designed to preserve stable input representations and training dynamics, which is crucial for the practical utility of DRL agents (Sutton, 2025). The design of mainstream intervention strategies is primarily motivated by four intuitions (Klein et al., 2024): weight perturbation, weight regularization, activation control, and training control. Weight perturbation (Ash & Adams, 2020; Elsayed et al., 2024; Hernandez-Garcia et al., 2024) involves directly clipping or perturbing the parameter weights of the policy to mitigate the adverse effects of outlier weights on the training dynamics. Weight regularization (Gogianu et al., 2021; Lyle et al., 2022; Dohare et al., 2024) applies soft constraints on weight updating to encourage the exploration of the policy in parameter space. Activation control (Lyle et al., 2023; Sokar et al., 2023; Nikishin et al., 2022; Abbas et al., 2023) involves regulating intermediate activations (e.g., normalizing activations and modifying activation functions) to reduce the sensitivity of representations to non-stationary inputs. Training control (Lee et al., 2023; Nikishin et al., 2023; Lee et al., 2024; Ma et al., 2024a) modifies the optimization process or objective to steer policy updates in a more stable direction, reducing the risk of aggressive updates that could lead to suboptimal solutions due to environmental dynamics. In addition, recent studies suggest that combining different interventions has the potential to further reduce the plasticity loss of the policy (Lyle et al., 2024a;b).

## 3 PROBLEM FORMULATION

**Threat Model.** The attack scenario involves a provider and an adversary (see Figure 2). The provider trains the DRL agent from scratch for the benign task and determines whether interventions should be applied to improve the agent's continual learning capability. Then, the provider deploys the well-trained agent or uploads it to a third-party platform. Based on the stage at which the adversary initiates the backdoor injection, we consider two widely discussed threat models (Kiourti et al., 2020; Cui et al., 2024; Rathbun et al., 2024; Ma et al., 2025; Dai et al., 2025):

*TM-Scratch.* The adversary injects the backdoor while the provider is training the agent.

*TM-Post.* The adversary downloads the agent released by the provider, injects a backdoor via post-training, and then republishes the backdoored agent to the third-party platform.

During the deployment phase, the backdoored agent performs sequential decision-making normally on the benign task. However, when the adversary inserts a trigger into the environment, the backdoor is activated, forcing the agent to output the target action. Appendix B provides supplementary clarification of the threat model and a detailed description of the backdoor injection.

---

[2]The source code is available at https://anonymous.4open.science/r/plasticity

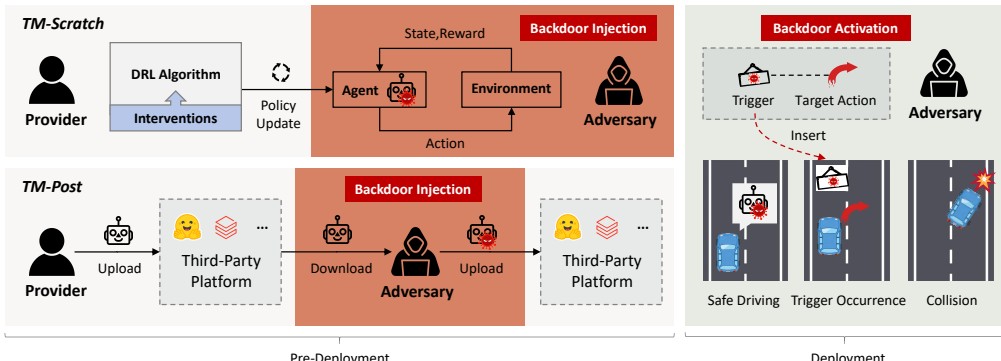

Figure 2: Threat models.

**Formulation.** The benign task is modeled as a Markov Decision Process, denoted by $\mathcal{M} = (\mathcal{S}, \mathcal{A}, \mathcal{R}, \mathcal{P}, \gamma)$, where $\mathcal{S}$ is the state space, $\mathcal{A}$ is the action space, and $\mathcal{R}$ is the reward function. The state transition function $\mathcal{P} : \mathcal{S} \times \mathcal{A} \to \Delta(\mathcal{S})$ defines the probability of reaching state $s' \in \mathcal{S}$ after taking action $a \in \mathcal{A}$ in state $s \in \mathcal{S}$. The discount factor $\gamma \in [0, 1)$ balances immediate and future rewards. The policy of the agent $\pi_\theta : \mathcal{S} \to \Delta(\mathcal{A})$ maps each state to a probability distribution over actions. The DRL algorithm aims to find the optimal parameters for the policy $\pi_\theta$ by maximizing the expected cumulative reward, i.e., $\theta^* = \arg\max_\theta \mathbb{E}_{\pi_\theta}[\sum_{t=0}^{T} \gamma^t r_t]$, where $T$ is the time horizon.

The backdoor task is modeled as $\mathcal{M}^\dagger = (\mathcal{T}, \mathcal{S}^\dagger, \mathcal{A}^\dagger, \mathcal{F}_s, \mathcal{F}_a, \mathcal{R}^\dagger)$, where $\mathcal{T}$ is the trigger space, $\mathcal{S}^\dagger \subseteq \mathcal{S}$ is the backdoor state space, and $\mathcal{A}^\dagger \subseteq \mathcal{A}$ is the target action space. $\mathcal{F}_s : \mathcal{S} \times \mathcal{T} \to \mathcal{S}^\dagger$ is a trigger-state mapping function that defines how a trigger alters a benign state. $\mathcal{F}_a : \mathcal{T} \to \mathcal{A}^\dagger$ is a trigger-action mapping function specified by the adversary, which determines the target action $a^\dagger \in \mathcal{A}^\dagger$ corresponding to each trigger $\delta \in \mathcal{T}$. $\mathcal{R}^\dagger$ is a backdoor reward function crafted to reinforce the mapping between triggers and target actions. The objective of the backdoor task is for the agent to output an action as close as possible to the target action when the backdoor is activated, i.e., $\min_{\theta^\dagger} \mathbb{E}_{s \sim \mathcal{S}, \delta \sim \mathcal{T}} \left[ ||\pi_{\theta^\dagger}(\mathcal{F}_s(s, \delta)) - a^\dagger|| \right]$, where $\pi_{\theta^\dagger}$ is the backdoored policy. Meanwhile, the adversary seeks to avoid degrading the agent's performance on the benign task.

## 4  STUDY DESIGN

**Attack Scenarios.** We construct diverse attack scenarios to underpin a comprehensive investigation. • For DRL tasks (i.e., benign tasks), we adopt four classic control tasks (CartPole, Acrobot, MountainCar, and Pendulum) and two physics control tasks (Lunar Lander and BipedalWalker) from OpenAI Gym (Brockman et al., 2016), as well as three robotic tasks (Hopper, Reacher, and HalfCheetah) from Facebook AI's PyBullet (Coumans & Bai, 2021). These tasks span discrete and continuous action spaces, sparse and dense reward structures, and both cold-start and non-cold-start conditions. • We consider performing backdoor injection during both the learning-from-scratch and post-training stages (i.e., *TM-Scratch* and *TM-Post*). • The backdoor attacks are carried out using four representative methods, all of which are compatible with both *TM-Scratch* and *TM-Post*: TrojDRL (Kiourti et al., 2020), BadRL (Cui et al., 2024), SleeperNets (Rathbun et al., 2024), and UNIDOOR (Ma et al., 2025). • We construct 47 backdoor tasks with reference to Ma et al. (2025), including both single-backdoor and multi-backdoor injection settings.

**Intervention Setup.** We consider eight intervention settings: • *None*: no intervention is applied, serving as a baseline for comparison; • *Shrink & Perturb* (Ash & Adams, 2020): upon each network update, weights are scaled by a small scalar and perturbed by adding weights from a randomly initialized network; • *Weight Clipping* (Elsayed et al., 2024): constraining the weights to lie within a predefined range; • *Spectral Normalization* (Gogianu et al., 2021): applied after the initial linear layer of the network; • *Weight Decay* (Dohare et al., 2024): setting the $\ell_2$ penalty coefficient to $10^{-5}$; • *Layer Normalization* (Lyle et al., 2023): applied after every linear layer; • *ReDo* (Sokar et al., 2023): periodically resetting the neurons with the highest dormancy level in each layer at fixed intervals; • *SAM* (Lee et al., 2023): applied Sharpness-Aware Minimization with Adam as the base optimizer, setting the sharpness penalty to 0.01. We formalize these intervention settings as a set $P = \{p_1, p_2, \ldots, p_8\}$, where each $p_i$ corresponds to the $i$-th intervention settings introduced

above (e.g., $p_8$ denotes *SAM*). We also consider two existing combinations, *Swiss Cheese* (Lyle et al., 2024a) and *Plastic* (Lee et al., 2023), alongside three newly introduced combinations: *Lac*, *SLac*, and *SSW*. Appendix C provides the implementation details of these combinations.

**Pathological Characteristics.** Existing studies suggest that interventions influence a DRL agent's continual learning capability through three pathological characteristics: weight magnitude, effective rank, and loss landscape sharpness (Lyle et al., 2023; Sokar et al., 2023; Dohare et al., 2024; Klein et al., 2024). In line with these studies, we analyze how interventions affect DRL backdoor attacks through these pathologies. Appendix D provides the conceptual definitions of these pathologies and details on their quantification. We formalize these pathologies as a set $C = \{c_1, c_2, c_3\}$, where each $c_i$ corresponds to the $i$-th pathology introduced above (e.g., $c_1$ denotes weight magnitude).

**Evaluation Metrics.** Consistent with prior studies (Li et al., 2025; Ma et al., 2025; Dai et al., 2025), we employ Attack Success Rate (ASR) and Benign Task Performance (BTP) as our primary evaluation metrics, measuring the attack's effectiveness and stealth, respectively:

$$\text{ASR} = \frac{1}{N_o} \sum_{i=1}^{N_o} \mathbf{1}\left[\pi_{\theta^\dagger}(\mathcal{F}_s(s_i, \delta_i)) = \mathcal{F}_a(\delta_i)\right], \tag{1}$$

where $N_o$ is the number of trigger occurrences, and $\mathbf{1}[\cdot]$ is the indicator function. In continuous action scenarios, since the output actions may not exactly coincide with the target actions, the indicator function is replaced with $\mathbf{1}[||\pi_{\theta^\dagger}(\mathcal{F}_s(s_i, \delta_i)) - \mathcal{F}_a(\delta_i)|| \leq \epsilon]$, where $\epsilon$ is the tolerance threshold.

$$\text{BTP} = \text{clip}(\frac{1}{N_e} \sum_{i=1}^{N_e} \frac{\sum_{t=0}^{T} \mathcal{R}(s_t, \pi_{\theta^\dagger}(s_t)) - B_l}{B_u - B_l}, 0, 1), \tag{2}$$

where $N_e$ is the number of episodes evaluated, $B_l$ denotes the expected return of a random policy on the benign task, and $B_u$ denotes the target performance of the benign task.

Appendix E provides additional design details, such as DRL algorithm implementations, backdoor attack implementations, and specific backdoor task designs.

## 5 EMPIRICAL STUDY AND ANALYSIS

This section mainly consists of three parts: (1) we investigate RQ1 by comparing the impact of eight intervention settings; (2) we probe RQ2 through the lens of three pathological characteristics; and (3) we explore RQ3 from the two dimensions of internal properties and external manifestations.

### 5.1 RQ1: IMPACT OF INTERVENTIONS

The results reported in this part cover 9,024 cases (2 threat models × 8 intervention settings × 47 backdoor tasks × 4 backdoor attacks × 3 random seeds).

Figure 3 shows that in *TM-Scratch*, interventions exert modest impact on DRL backdoor attacks. Specifically, figure 3(a) shows that ASR exhibits minor fluctuations, with the most pronounced intervention being *Layer Normalization* at -8.84%. Figure 3(b) shows that *Spectral Normaliza-*

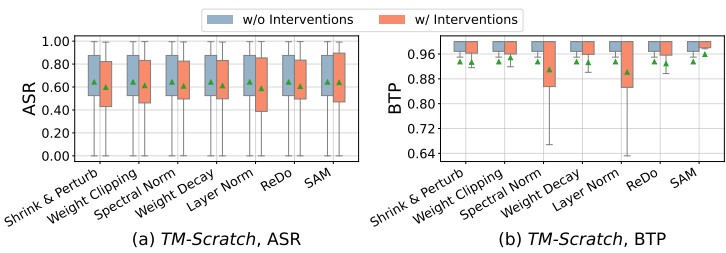

(a) *TM-Scratch*, ASR      (b) *TM-Scratch*, BTP

Figure 3: Impact of interventions in *TM-Scratch*.

*tion* and *Layer Normalization* cause relatively pronounced fluctuations in BTP. For instance, *Layer Normalization* reduces BTP by over 30% in the Acrobot and MountainCar environments, both of which are characterized by sparse rewards. Among them, *SAM* is the only intervention that slightly improves BTP while leaving ASR virtually unchanged. Specific numerical results can be found

in Table 6 in the Appendix. Furthermore, Appendix F shows that these interventions have negligible impact on BTP under conventional training. Therefore, the fluctuations in BTP observed in Figure 3(b) are caused by the effects of interventions on backdoor attacks.

Figure 4 shows that in *TM-Post*, interventions have a more pronounced impact on DRL backdoor attacks, which aligns with expectations since interventions affect a well-trained DRL agent (suffering from plasticity loss) more than a randomly initialized agent. Figure 4(a) shows that most

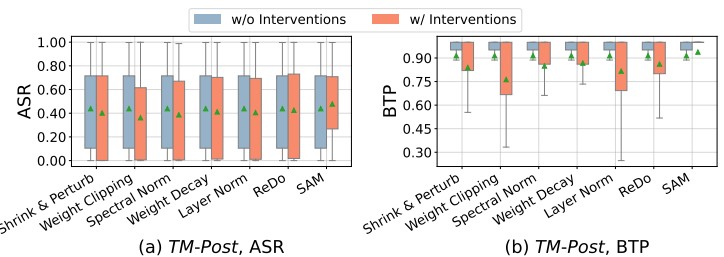

(a) *TM-Post*, ASR        (b) *TM-Post*, BTP

Figure 4: Impact of interventions in *TM-Post*.

interventions reduce ASR, with *Weight Clipping* decreases it by an average of 17.46% and *Spectral Normalization* decreases it by an average of 11.78%. Figure 4(b) shows that most interventions significantly reduce BTP, with *Weight Clipping* decreases it by an average of 20.19% and *Layer Normalization* decreases it by an average of 11.93%. Remarkably, we find that in *TM-Post*, *SAM* produces a pronounced enhancement of backdoor attacks. For instance, in robotic tasks, it increases the ASR from 0.178 to 0.326, a 83.15% relative improvement, while in physics control tasks, the gain is 20.68%. Specific numerical results can be found in Table 7 in the Appendix.

> **Finding 1**: In *TM-Post*, interventions exert a more substantial impact on DRL backdoor attacks than in *TM-Scratch*. Notably, *SAM* exacerbates the backdoor threat, whereas the other interventions exhibit varying degrees of mitigation.

### 5.2 RQ2: INTRINSIC MECHANISMS

In this part, we first examine the impact of backdoor attacks on the agent with respect to the three pathological characteristics, establishing a baseline for subsequent comparison with interventions. Figure 5 shows that backdoor attacks further increase the non-stationarity of DRL training, as reflected by larger performance oscillations. The performance ranges (i.e., the absolute differences between the maximum and minimum results) of weight magnitude and effective rank increase by 98.63% and 19.16%, respectively, with the most pronounced effect observed in loss landscape sharpness, whose range increases by 635.22%.

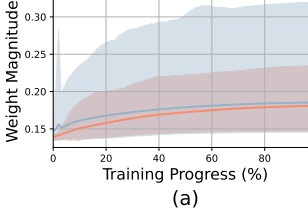 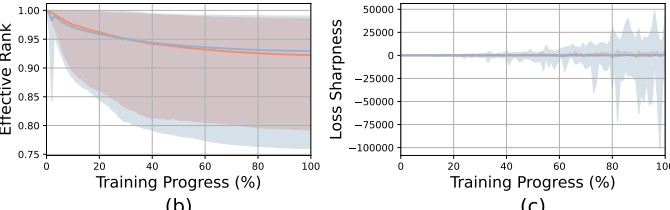

Figure 5: Comparison of conventional training and backdoor attacks on the three pathological characteristics. Solid lines represent mean values, and shaded areas denote the range from minimum to maximum values. Legend representation: — Conventional Training, — Backdoor Attacks.

Then, we monitor the effects of interventions across these pathologies and rank them accordingly. For any intervention setting $p_i \in P$, we define a pathological vector $\mathbf{v}(p_i) = (v_{i1}, v_{i2}, v_{i3})$, where $v_{ij} \in [1, 8]$ represents the measurement of the agent on the pathological characteristic $c_j \in C$ under $p_i$.

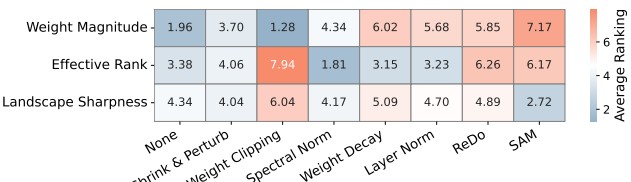

Figure 6: Impact of interventions on backdoor attacks across three pathological characteristics.

Appendix G discusses the motivation for ranking and provides the ranking criteria. Figure 6 presents

the average ranking results. For example, $\mathbf{v}(p_2) = \big(3.70, 4.06, 4.04\big)$ summarizes the overall status of *Shrink & Perturb* across the three pathologies, where $v_{21} = 3.70$ indicates that the backdoored agent achieves an average ranking of 3.70 on weight magnitude across all cases after applying *Shrink & Perturb*. More details on the ranking results are provided in Figure 14 of the Appendix.

**Weight Magnitude.** Figure 6 shows that on the weight magnitude dimension, only $v_{31} < v_{11}$, indicating that *Weight Clipping* is the sole intervention that reduces the weight magnitude of the backdoored agent, whereas all other interventions result in varying degrees of increase. Further, we record the weight magnitude of the second linear layer in the agent's actor network. Figure 7(a) corresponds to conventional training situation, and Figure 7(b) to a backdoored agent without interventions. The results show that backdoor attacks lead to a significant increase in the magnitude of certain weights (see red boxes). This indicates that the weights strongly associated with the backdoor are sparse, making the backdoor pathways more fragile than those supporting benign tasks.



Figure 7: Visualization of the weight magnitude in the actor network's second fully connected layer.

As shown in Figure 7(c), *Weight Clipping* clips all weights exceeding the threshold, thereby permanently constraining them within predefined bounds. Its intrinsic mechanism is that clipping disrupts both backdoor and benign pathways, inducing reconstruction competition that exacerbates nonstationarity in DRL training and degrades performance. This resembles certain mitigation strategies proposed in the deep learning backdoor domain (Li et al., 2024b). *Weight Clipping* has limited impact on backdoor attacks in *TM-Scratch*, primarily because (1) the overall weight magnitudes are relatively small, resulting in only a few weights being clipped, and (2) the actor network possesses sufficient parameter flexibility to rapidly reconstruct both benign and backdoor pathways after each clipping. However, these properties no longer hold in *TM-Post*, resulting in the suppression of backdoor attacks. Figure 15 in the Appendix presents this contrast through a 3D visualization.

Because backdoor pathways are sparse, interventions that share intrinsic mechanisms with *Weight Clipping* do not necessarily achieve high rankings in the weight magnitude dimension, such as *Shrink & Perturb* and *ReDo*. *Shrink & Perturb* periodically applies mild compression to all weights, followed by the injection of small perturbations. *ReDo* resets a small subset of neurons that are dormant with respect to benign tasks, which may inadvertently disrupt backdoor-associated neurons. As shown in Figure 7, neurons that are strongly associated with backdoors tend to exhibit only weak relevance to benign tasks. These two interventions disrupt backdoor pathways in a softer manner, resulting in limited competition during pathway reconstruction, and therefore only produce mild mitigation of backdoor attacks.

> **Finding 2**: Interventions characterized by noise, clipping, and reset (e.g., *Shrink & Perturb*, *Weight Clipping*, and *ReDo*) disrupt activation pathways, leading to competitive reconstruction between backdoor and benign pathways.

**Effective Rank.** Figure 6 shows that *Spectral Normalization* achieves the highest average ranking in the effective rank dimension (i.e., $v_{42} = \min_{i \in P} v_{i2} = 1.81$). *Spectral Normalization* achieves this by normalizing each weight matrix with its largest singular value, thereby constraining

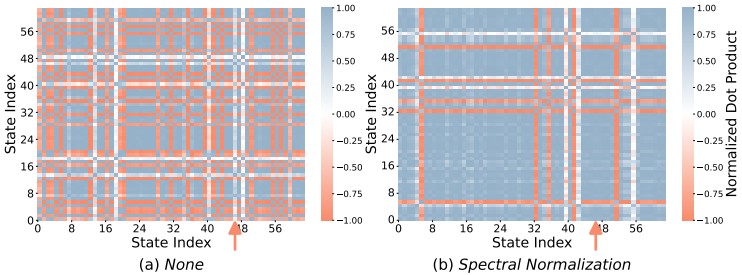

Figure 8: Normalized dot product of the actor network's gradients over 64 states. The red arrow marks the backdoor state.

the Lipschitz constant of the actor network and effectively compressing the agent's representation space. This process implicitly enhances the relative importance of smaller singular values, allowing more directions in the parameter space to contribute to representation, which reduces the prominence of the trigger. To provide a more intuitive understanding of the above effects, we compute the normalized dot product (Lyle et al., 2023) of actor network's gradients over 64 states, with implementation details provided in Appendix H.

Figure 8 shows a typical case, where the red arrow marks the backdoor state (state index = 46). Figure 8(a) shows that, in the absence of interventions, the gradient direction of the backdoor state is close to 0 relative to other benign states, indicating that the backdoor and benign tasks exhibit orthogonality (Zhang et al., 2024). Figure 8(b) shows that the gradient directions approach 1, indicating that *Spectral Normalization* is capable of aligning backdoor gradients with those of benign states. Consequently, the backdoor pathways become dense rather than sparse, meaning they overlap more with benign pathways and form shared pathways. These shared pathways are harder to stabilize during non-stationary DRL training, resulting in fluctuations in backdoor attack performance.

*Weight Decay* and *Layer Normalization* also increase the effective rank (i.e., $v_{52} < v_{12}$ and $v_{62} < v_{12}$), and their impact on backdoor attacks follows an intrinsic mechanism similar to that of *Spectral Normalization*. *Weight Decay* constrains the distances between weights, guiding the backdoor representation to associate with more weights in a soft manner. *Layer Normalization* reduces internal covariate shift by normalizing activations within each layer, effectively smoothing the actor network's response to input perturbations, thereby making the representation of the backdoor state closer to that of benign states (see Figure 16 in the Appendix).

> **Finding 3**: Interventions that compress the representation space shift the gradient directions of backdoor and benign states from orthogonal to aligned (e.g., *Spectral Normalization*, *Weight Decay*, and *Layer Normalization*), transforming backdoor pathways from sparse to dense.

**Loss Landscape Sharpness.** As illustrated in Figure 5, excessive loss landscape sharpness emerges as a distinctive hallmark of backdoor attacks, compared with the other two pathological characteristics. This is because backdoor attacks introduce pronounced heterogeneity into the state distribution, with triggers functioning as artificially constructed rare signals. To establish the association between triggers and target actions from a limited number of transitions (typically assigned large backdoor rewards), DRL training induces sharp gradient changes in localized regions of the weight space, leading to a more peaked loss landscape.

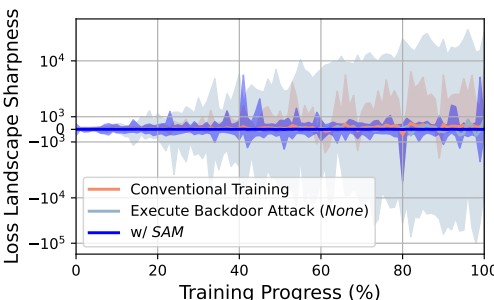

Figure 9: *SAM* flattens the loss landscape.

Figure 6 shows that *SAM* is the only intervention that significantly reduces the loss landscape sharpness of the backdoored agent (i.e., $v_{83} = \min_{i \in P} v_{i3} = 2.72$). Figure 9 further demonstrates that *SAM* compresses the loss landscape sharpness induced by backdoor attacks to an average of 2.11% across all cases. Counterintuitively, *SAM* does not mitigate backdoor attacks but instead amplifies them. Its intrinsic mechanism is that *SAM* captures the backdoor direction via the sharp losses (Foret et al., 2021; Zeng et al., 2025) and amplifies the corresponding gradients, enabling the backdoor pathway to rapidly converge into a flat-minimum region that is robust to parameter perturbations. This reduces continuous competition between backdoor and benign pathways in the inherently non-stationary training process of DRL. The effect is especially pronounced in *TM-Post*, where reduced flexibility in the agent's parameter space hinders the formation of the backdoor pathway. Appendix I presents a theoretical proof, leveraging influence functions (Koh & Liang, 2017) to demonstrate how *SAM* amplifies backdoor threats. Figure 17 in the Appendix presents the attack performance distributions of *SAM* and other intervention settings in the form of contour plots.

> **Finding 4**: *SAM* amplifies the backdoor gradients and enables the backdoor pathway to rapidly converge while remaining robust to parameter perturbations, thereby alleviating competition between backdoor and benign tasks—a phenomenon especially pronounced in *TM-Post*.

**Remarks.** This part serves as a supplement to the cause analysis presented in this section: (1) In *TM-Scratch*, the representations of benign and backdoor tasks are competitively co-constructed, so the effects of interventions on yet-to-be-stabilized representations are continuously reshaped and diluted by training dynamics. In *TM-Post*, the agent has already established representations corresponding to the benign task. Injecting a backdoor at this stage requires forcibly carving out pathways in the existing weights, which intensifies the competitive conflict between backdoor and benign tasks. Consequently, interventions exert a more pronounced impact on DRL backdoor attacks in *TM-Post*. (2) Benign representations involve complex decision-making and rely on the coordinated activity of a large number of network parameters. Interventions typically restrict parameter flexibility, making it more difficult to reconstruct disrupted benign representations. In contrast, backdoor representations often require only a small number of parameters or localized pathways. Even under constrained parameter flexibility, backdoor representations are rapidly reconstructed. As a result, the influence of interventions on BTP is generally more pronounced than on ASR.

## 5.3 RQ3: IMPLICATIONS FOR FUTURE RESEARCH

Since the impacts of interventions on DRL agents arise from different intrinsic mechanisms, numerous studies have demonstrated that appropriately combining different interventions yields additive effects for maintaining plasticity. Therefore, we investigate what novel effects on

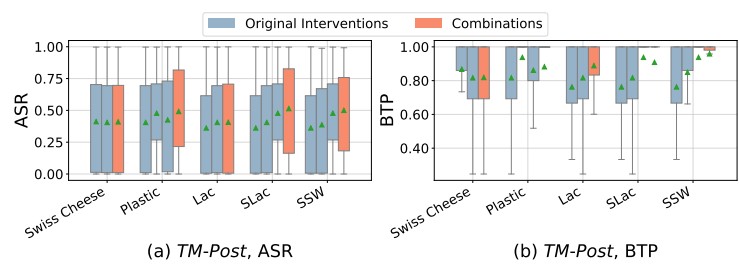

(a) *TM-Post*, ASR      (b) *TM-Post*, BTP

Figure 10: Impact of combinations in *TM-Post*.

DRL backdoor attacks emerge when these interventions are combined. The results reported in this part cover 5,640 cases. Specific numerical results can be found in Tables 8 and 9 in the Appendix.

Figure 10 shows that the mitigative effect of *Swiss Cheese* is nearly identical to that of *Layer Normalization* alone, whereas *Lac* exhibits a less effective mitigation than the two original interventions. This suggests that the mitigative effects of interventions on backdoor attacks are non-additive. Counterintuitively, we find that

Table 1: Combination impacts of *SAM*.

| Combinations | ASR | BTP |
|---|---|---|
| *Plastic* | +106.74% | -2.82% |
| *SLac* | +134.27% | +9.53% |
| *SSW* | +134.83% | +22.82% |

combining these interventions with *SAM* in *TM-Post* may further amplify backdoor threats compared with using *SAM* alone. For instance, Table 1 shows that in robotic tasks, the ASR gains of *Plastic*, *SLac*, and *SSW* increase progressively, reaching up to 134.83%.

> **Finding 5**: In *TM-Post*, interventions whose intrinsic mechanisms involve disrupting activation pathways and compressing the representation space may act as catalysts for DRL backdoor attacks when combined with *SAM*.

**Robust Backdoor Injection.** Motivated by the above finding, we propose a conceptual framework, *SCC* (see Figure 11), for facilitating robust backdoor attacks in *TM-Post*. *SCC* alters the internal properties of DRL backdoors and comprises three components: *Scavenger, Converter Connector*.

- *Scavenger* releases a subset of benign pathways in the well-trained DRL agent, enabling the subsequent construction of backdoor pathways. Its design can draw upon interventions such as *Shrink & Perturb*, *Weight Clipping*, and *ReDo*.
- *Converter* aligns backdoor and benign gradients, conferring multi-pathway characteristics to the backdoor and addressing the vulnerability limitations of sparse backdoor pathways. Its design can draw upon interventions such as *Spectral Normalization*, *Weight Decay*, and *Layer Normalization*.
- *Connector* ensures stable joint construction of backdoor and benign representations across multiple pathways. Its design can draw upon interventions such as *SAM*.

Under the non-stationary dynamics of DRL, multi-pathway representations exhibit greater robustness than single pathways, accounting for the heightened backdoor threat posed by *Plastic*, *SLac*,

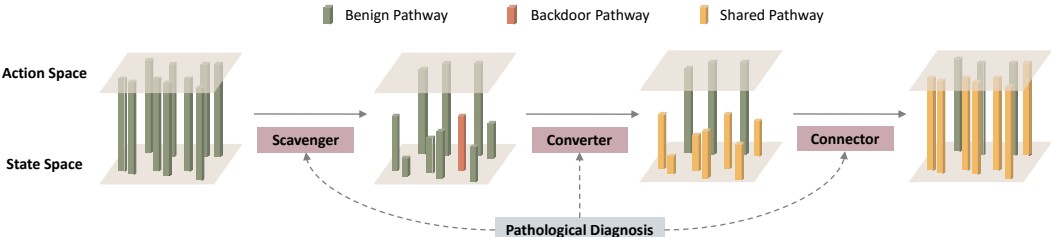

Figure 11: The SCC framework.

and *SSW* relative to *SAM*. To elucidate the causes of performance variations among the three combinations, we propose a novel notion, *Pathological Diagnosis (PD)*, which quantifies the pathological distances among interventions in a combination. Specifically, *Pathological Diagnosis* consists of two steps: (1) *Compute Pairwise Distance*: The pairwise distance between two interventions is defined as the Euclidean distance between their pathological vectors:

$$d(p_i, p_j) = \|\mathbf{v}(p_i) - \mathbf{v}(p_j)\|_2. \tag{3}$$

(2) *Compute Pathological Distance*: For a combination $A$, the pathological distance is defined as the sum of pairwise distances among all interventions in $A$:

$$PD(A) = \sum_{1 \leq i < j \leq |A|} d(p_i, p_j), \tag{4}$$

where $|A|$ denotes the number of combined interventions in $A$. For instance, *Swiss Cheese* comprises *Weight Decay* and *Layer Normalization*, with $\mathbf{v}(p_5) = (6.13, 3.11, 5.09)$ and $\mathbf{v}(p_6) = (5.73, 3.18, 4.51)$, yielding a pathological diagnosis result of $PD(Swiss\ Cheese) = 0.71$. Then, we obtain $PD(Plastic) = 9.43$, $PD(SLac) = 17.42$, and $PD(SSW) = 18.64$. This implies that increasing the pathological distances among the three components facilitates the amplification of backdoor threats, as they minimally interfere with each other at the pathological level.

**Sharpness-Based Detection.** We find that abnormal loss landscape sharpness emerges as a salient external manifestation of DRL backdoor attacks (as shown in Figure 5(c)). With the exception of *SAM*, most interventions exacerbate this phenomenon by rendering the loss landscape even sharper (e.g., $v_{33} > v_{13}$). These observations highlight sharpness-based detection as a promising direction for future exploration. For instance, a defender monitoring sharpness in real time throughout the agent's training may detect abnormal spikes or drops that signal potential backdoor threats. Two challenges merit further investigation: first, sharpness exhibits substantial variation across different DRL tasks, complicating the establishment of a unified detection threshold; second, other factors that could induce abnormal sharpness remain insufficiently understood, and disentangling these sources is critical for reducing false positives.

## 6 CONCLUSION

This study investigates the impacts of plasticity interventions on existing DRL backdoor attacks. We empirically study 14,664 representative cases and distill these impacts into three intrinsic mechanisms. Our findings show that integrating these mechanisms changes the internal properties of DRL backdoors and makes them more robust, leading us to propose the conceptual framework *SCC* for the post-training scenario. In addition, from the perspective of external manifestation, we propose that sharpness-based detection may be a promising direction for future research. Overall, this study seeks to promote the consideration of potential threats in tandem with DRL advancements, thereby contributing to the foundation for secure deployment.

ETHICS STATEMENT

**Stakeholder Analysis.** We conduct a comprehensive stakeholder analysis to identify the primary parties engaged in researching backdoor threats in DRL systems. These stakeholders include research institutions, universities, companies, and practitioners who are actively involved in advancing DRL for cutting-edge scientific problems and real-world applications. For each group, we carefully evaluate their potential interests as well as the associated risks.

**Potential Outcomes.** This study aims to raise awareness among institutions and individuals dedicated to advancing DRL research and societal progress about the latent risks posed by backdoor threats. At the same time, it encourages practitioners to not only pursue performance improvements in DRL systems but also consciously consider the security implications of techniques such as plasticity interventions. This, in turn, can drive the development of more robust countermeasures. As the attack pipeline for DRL backdoors represents an objective reality, ignoring its potential threats is futile. Instead, directly confronting these challenges and mitigating the associated risks constitutes the core motivation of this study.

**Responsible Dissemination.** Consistent with our commitment to ethical research, we plan to disseminate the findings and code associated with this study responsibly. Alongside the open-source release, we will include a statement addressing the ethical considerations surrounding this work.

REPRODUCIBILITY STATEMENT

We recognize the critical role of transparency and reproducibility in research. Therefore, we provide a link to the source code of this study, which includes the implementations of plasticity interventions, combination strategies, and pathological characteristics. During the review process, this link remains anonymized to comply with double-blind review requirements.

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

# APPENDIX

## CONTENTS

## THE USE OF LARGE LANGUAGE MODELS

Our use of LLMs encompasses two aspects: (1) ChatGPT (OpenAI) is used for grammar checking and language refinement. (2) Cursor (Anysphere) assists coding by debugging and visualizing data. In all cases, the LLMs served solely as auxiliary tools. Any content generated by LLMs served solely as raw material, which the authors carefully reviewed, modified, and validated as needed. All ideas, conceptualization, and primary content of the paper are created solely by the authors.

## A  FURTHER BACKGROUND

DRL is driving advances across various domains and has been widely adopted in security research (Wilson et al., 2025; Qiao et al., 2024; Xia et al., 2023). However, despite these advances, DRL faces numerous potential security challenges, including adversarial attacks, poisoning attacks, and issues related to copyright protection.

**Adversarial Attacks.**  The most straightforward form of adversarial attack involves adding perturbations to the environment or the observations, thereby disrupting the victim agent's sequential decision-making (Behzadan & Munir, 2017; Huang et al., 2017; Sun et al., 2020; Tu et al., 2021). Such approaches draw inspiration from adversarial examples in deep learning (Carlini & Wagner, 2017). Another novel class of attacks is adversarial policies (Gleave et al., 2020; Guo et al., 2021; Wang et al., 2023; Ma et al., 2024b), which exploit the tendency of DRL algorithms to overfit and

the lack of Nash equilibrium guarantees in competitive environments to rapidly uncover vulnerabilities in the victim agent's policy. Such attacks can be used not only to achieve indirect manipulation of actions but also to evaluate robustness lower bounds.

**Poisoning Attacks.** Compared to single-step decision systems, altering the long-term objectives of sequential decision-making systems through poisoning is more challenging. Existing studies (Mohammadi et al., 2023; Li et al., 2024a) demonstrate that the essence of poisoning attacks lies in maliciously manipulating the reward function or transition data to steer the agent away from its intended objectives and induce policy updates toward an adversary-predefined goal. Such attack techniques have also been extended to safety alignment (Baumgärtner et al., 2024; Pathmanathan et al., 2025; Betley et al., 2025) and are often employed as an effective means to inject DRL backdoor attacks.

**Copyright Protection.** With the growing practical applications of DRL, the issue of copyright protection has increasingly attracted attention. Adversaries stealing policy networks (Chen et al., 2021b) may trigger copyright disputes, while watermarking techniques (Chen et al., 2021a) can partly mitigate this issue. In addition, both training environments and hyper-parameters in DRL pose risks of privacy leakage (Pan et al., 2019; Du et al., 2025). Moreover, online DRL paradigm relies on interaction experiences with the environment, which assigns intrinsic value to the environment itself. Reinforcement unlearning techniques (Ye et al., 2025) can selectively remove the learned knowledge of the training environment from the agent's memory. Offline DRL paradigm relies on expert-generated trajectory data, where trajectory-level auditing mechanisms (Du et al., 2024) and trajectory unlearning techniques (Gong et al., 2025) can be employed to enable copyright protection.

# B SUPPLEMENTARY INFORMATION ON THE THREAT MODEL

## B.1 CLARIFICATION OF THE THREAT MODEL

For the current *TM-Post* setting, two potential concerns may arise:

**Concern 1.** *In TM-Post, is the adversary required to strictly adhere to the interventions embedded by the provider?*

In *TM-Post*, the adversary is not necessarily constrained to follow the interventions embedded by the provider. For example, the adversary is allowed to remove interventions such as *Weight Clipping* or *ReDo*, which has a negligible impact on post-training. In contrast, the removal of interventions such as *Spectral Normalization* or *Layer Normalization* is generally infeasible, as it is prone to induce substantial performance degradation or even catastrophic failure of the DRL agent during post-training.

**Concern 2.** *If the adversary is not constrained to adhere to the provider's embedded interventions, does investigating this scenario remain meaningful?*

Considering *TM-Post* is essential, and it constitutes one of the primary motivations of this study. In existing studies, the adversary remains unaware of whether the interventions influence DRL backdoor attacks. Since one of the adversary's goals is to preserve the victim agent's performance on benign tasks (i.e., BTP), there is an incentive to retain the provider's embedded interventions, or even to introduce additional interventions to compensate for BTP degradation. Therefore, the adversary might overlook the fact that certain interventions could either exacerbate or mitigate the backdoor threat. This study provides insights for both the adversary and the provider/defender: the adversary leverages these insights to exacerbate backdoor threats, while the provider uses them to mitigate such threats.

## B.2 DETAILS OF BACKDOOR INJECTION

As illustrated in Figure 12, in a standard training pipeline, the DRL agent collects environmental information via sensors, where each dimension of the information is represented as a vector. These vectors are then concatenated to constitute the state. The agent inputs the state into the policy and obtains the action output. Following the execution of an action, the agent receives the reward signal from the environment. The state, action, and reward together constitute a transition. The agent stores these transitions and uses them to update the policy. In this context, executing a backdoor

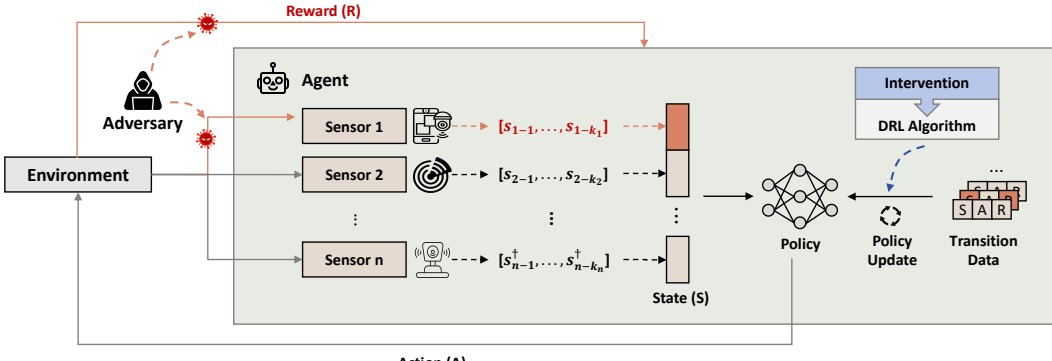

Figure 12: A conceptual illustration of backdoor injection.

injection requires the adversary to possess the capability to perturb both the state and reward. There are primarily two paradigms for accomplishing this:

**Paradigm 1.** The adversary introduces a trigger into the environment at low frequency, perturbing the agent's perception of specific environmental dimensions within a predefined range. The adversary monitors the agent's action outputs and then perturbs the reward signal (typically increasing it) when the outputs match the predefined target action, thereby compelling the agent to learn the mapping between the trigger and the target action.

**Paradigm 2.** The adversary have the authority to tamper with the transitions stored by the agent. In such cases, the adversary directly tampers with a small portion of the transitions (e.g., less than 1%) by modifying the state and reward. This is sufficient to force the agent to associate the trigger with the target action. Some studies (Rathbun et al., 2025; Ma et al., 2025) also indicate that in continuous action scenarios, the target action may occur sparsely, in which case modifying the action along with the state and reward can further enhance the attack effectiveness.

## C COMBINATION STRATEGIES

In this study, we consider two representative combination strategies: *Swiss Cheese* (Lyle et al., 2024a) and *Plastic* (Lee et al., 2023). *Swiss Cheese* combines *Weight Decay* and *Layer Normalization*, whereas *Plastic* integrates *Layer Normalization*, *SAM*, and *ReDo*. Furthermore, based on the results of RQ1 and the analysis of RQ2, we additionally investigate three new combinations: *Lac*, *SLac*, and *SSW*. *Lac* combines the two most mitigative interventions, *Weight Clipping* and *Layer Normalization*, to investigate whether their mitigating effects are additive. *SLac* extends *Lac* by incorporating *SAM*, aiming to explore how interventions with opposing effects on DRL backdoor attacks interact with each other. *SSW* combines interventions sorted highest across the three pathological characteristics (see Figure 6): *Weight Clipping*, which ranks highest in terms of weight magnitude (i.e., $v_{31} = \min_{i \in P} v_{i1} = 1.28$); *Spectral Normalization*, which ranks highest in terms of effective rank (i.e., $v_{42} = \min_{i \in P} v_{i2} = 1.81$); and *SAM*, which ranks highest in terms of loss landscape sharpness (i.e., $v_{83} = \min_{i \in P} v_{i3} = 2.72$). Table 2 lists the original interventions for all combinations discussed.

Table 2: Combinations and their interventions: ○ indicates exclusion, ● indicates inclusion.

| Combinations | Original Intervention Strategies | | | | | | |
|---|---|---|---|---|---|---|---|
| | *Shrink & Perturb* | *Weight Clipping* | *Spectral Norm* | *Weight Decay* | *Layer Norm* | *ReDo* | *SAM* |
| *None* | ○ | ○ | ○ | ○ | ○ | ○ | ○ |
| *Swiss Cheese* | ○ | ○ | ○ | ● | ● | ○ | ○ |
| *Plastic* | ○ | ○ | ○ | ○ | ● | ● | ● |
| *Lac* | ○ | ● | ○ | ○ | ● | ○ | ○ |
| *SLac* | ○ | ● | ○ | ○ | ● | ○ | ● |
| *SSW* | ○ | ● | ● | ○ | ○ | ○ | ● |

# D PATHOLOGY QUANTIFICATION

**Weight Magnitude.** We first accumulate the squared values of all weights in the linear layers, then compute the Root Mean Square (RMS) over all weights:

$$\text{Weight Magnitude} = \sqrt{\frac{1}{N} \sum_{l \in \mathcal{L}} \sum_{i,j} \left( W_{i,j}^{(l)} \right)^2},$$

where $\mathcal{L}$ is the set of linear layers, $W^{(l)}$ is the weight matrix of layer $l$, and $N = \sum_{l \in \mathcal{L}} \text{size}(W^{(l)})$ is the total number of weights across all linear layers.

**Effective Rank.** Given a weight matrix $W \in \mathbb{R}^{n \times m}$ (we take the penultimate linear layer), let its singular values be denoted as $\sigma_k$, where $k = 1, 2, \ldots, q$, and $q = \min(n, m)$. We define the normalized singular value distribution as $p_k = \frac{\sigma_k}{\|\sigma\|_1}$, where $\sigma = (\sigma_1, \ldots, \sigma_q)$ and $\| \cdot \|_1$ denotes the element-wise $\ell^1$-norm. Then, the effective rank is computed as:

$$\text{Erank}(W) = \exp\{H(p_1, p_2, \ldots, p_q)\},$$

where $H(p_1, p_2, \ldots, p_q) = -\sum_{k=1}^{q} p_k \log(p_k)$. To facilitate comparison across tasks with different hidden dimensions, we further define the effective rank ratio:

$$\text{Effective Rank Ratio} = \frac{\text{Erank}(W)}{d},$$

where $d$ denotes the hidden size of the corresponding layer.

**Loss Landscape Sharpness.** To quantify the sharpness of the loss landscape, we estimate the largest eigenvalue of the Hessian matrix with respect to model parameters using power iteration. We first flatten all trainable parameters of the model into a single vector $\theta \in \mathbb{R}^n$, where $n$ denotes the total number of parameters. A random vector $v \sim \mathcal{N}(0, I)$ is sampled from a standard multivariate Gaussian distribution and then normalized as $v \leftarrow v/\|v\|$.

At each iteration, we compute the Hessian-vector product $h_v = Hv$, where $H = \nabla_\theta^2 L(\theta)$ is the Hessian of the loss function. The Rayleigh quotient $\lambda = v^\top h_v$ serves as the current estimate of the dominant eigenvalue. The direction vector is then updated and re-normalized via

$$v \leftarrow \frac{h_v}{\|h_v\| + \varepsilon},$$

where $\varepsilon$ is a small constant to ensure numerical stability. After a fixed number of iterations, the final eigenvalue estimate $\lambda_{\max}$ is used to represent the loss landscape sharpness:

$$\text{Loss Landscape Sharpness} = \lambda_{\max}.$$

# E ADDITIONAL DESIGN DETAILS

**DRL Implementation.** The experiments are implemented in Python with PyTorch and conducted on a server equipped with 10 NVIDIA GeForce RTX 4090 GPUs. We adopt Proximal Policy Optimization (PPO) (Schulman et al., 2017), one of the most widely used DRL algorithms, which is a policy gradient method that optimizes a stochastic policy with importance sampling and a clipped objective function to enhance training stability. PPO follows the actor-critic architecture, where the critic network is parameterized by a 3-layer MLP with hidden size 64 and Tanh activations. For the actor network, a 3-layer MLP is used for tasks with discrete action spaces, while a 4-layer MLP is applied to tasks with continuous action spaces, both with hidden size 64 and Tanh activations. Orthogonal initialization with standard deviation $\sqrt{2}$ is applied to the weights, and biases are initialized to 0. The networks are trained using the Adam optimizer. Following Raffin et al. (2021), hyper-parameters such as learning rate and batch size are configured individually for each DRL task.

**Backdoor Attack Implementation.** We incorporate an action tampering module into all attacks to mitigate the issue of low target-action occurrence frequency. For all backdoor attacks, the poisoning rate is capped at 0.4%, and the tolerance threshold $\epsilon$ is set to 0.1 for Bipedal Walker and 0.05 for the other tasks. To ensure backdoor task alignment across all attack methods, we remove the trigger

optimization component from BadRL. In SleeperNets, the reward constant is fixed at 5 and the weighting factor at 0.5.

**Benign Task Selection.** We select 9 benign tasks that span five key dimensions, capturing diverse characteristics of DRL environments. Specifically, they differ in the type and dimensionality of actions (discrete vs. continuous, one-dimensional vs. multi-dimensional), the nature of the reward signal (sparse vs. dense, w/ or w/o normalization), and whether the task involves a cold-start challenge. Table 3 summarizes these tasks.

Table 3: Summary of benign tasks used for investigation.

| Categories | DRL Tasks | Action Space | Action Dim. | Reward Type | Reward Norm | Cold-Start |
|---|---|---|---|---|---|---|
| **Classic Control Tasks** | CartPole | Discrete | 1D | Dense | × | × |
| | Acrobot | Discrete | 1D | Sparse | × | × |
| | MountainCar | Discrete | 1D | Sparse | × | ✓ |
| | Pendulum | Continuous | 1D | Dense | × | × |
| **Physics Control Tasks** | Lunar Lander | Discrete | 1D | Dense | × | × |
| | Bipedal Walker | Continuous | N-D | Dense | × | ✓ |
| **Robotic Tasks** | Hopper | Continuous | N-D | Dense | ✓ | × |
| | Reacher | Continuous | N-D | Dense | ✓ | × |
| | Half Cheetah | Continuous | N-D | Dense | ✓ | ✓ |

**Backdoor Task Design.** Table 10 presents the specific settings of the 47 backdoor tasks. Specifically, we define a backdoor unit as a triplet $(s^{(d)}, \delta^{(v)}, a^{\dagger})$, where $s^{(d)}$ specifies the state dimension into which the trigger is injected, $\delta^{(v)}$ indicates the specific perturbation applied to that state dimension, and $a^{\dagger}$ denotes the target action associated with the trigger. The pair $(s^{(d)}, \delta^{(v)})$ together constitute an independent trigger $\delta \in \mathcal{T}$, as described in Section 3. These tasks can be categorized into single-backdoor tasks and multi-backdoor tasks:

- A single-backdoor task contains one backdoor unit. For example, `Task1` specifies that the adversary in CartPole perturbs the 0-th state dimension to -4.8, thereby forcing the backdoored agent to output the target action `push cart to the right`.
- A multi-backdoor task contains multiple backdoor units. For example, `Task32` specifies that the adversary in Lunar Lander can perturb the 0-th state dimension to -1.5, causing the backdoored agent to output the target action `do nothing`, or perturb the 4-th state dimension to 3.14, causing the agent to output the target action `fire main engine`.

**Trigger Examples.** To facilitate an intuitive understanding of the trigger $(s^{(d)}, \delta^{(v)})$ in a backdoor unit, we provide four illustrative examples (as shown in Figure 13):

- $(s^{(6)}, 0)$ in `Task29`: In Lunar Lander, the adversary can perturb the lander's force sensor, causing it to output 0 and thereby affecting its judgment of whether the left leg has made contact with the ground.
- $(s^{(0)}, 3.14)$ in `Task30`: In Bipedal Walker, the adversary can perturb the walker's Inertial Measurement Unit (IMU), causing it to register the hull's angular velocity as reaching its maximum.
- $(s^{(1)}, 5)$ in `Task40`: In Hopper, the adversary can perturb the robot's IMU, causing it to register the body's angle relative to the x-axis as 5.
- $(s^{(2)}, -5)$ in `Task41`: In Half Cheetah, the adversary can perturb the robot's LiDAR, causing it to register the horizontal velocity of the torso as -5.

There exists a substantial body of prior work focused on attacking and perturbing such sensors Cao et al. (2019); Xu et al. (2023).

## F  IMPACT OF INTERVENTIONS ON CONVENTIONAL TRAINING

Since our analysis of interventions and combinations against DRL backdoor attacks involves benign task performance (BTP), we first examine their effects under conventional DRL training (i.e.,

Figure 13: Examples of trigger designs across four DRL tasks. From left to right: Lunar Lander, Bipedal Walker, Hopper, and Half Cheetah.

poisoning rate = 0.00%). Note that in this case, *None* is equivalent to conventional training. Table 4 reports the performance differences between various interventions and the baseline across all benign tasks, with values of +0.002, -0.004, -0.007, -0.006, -0.002, -0.012, and -0.010, and a standard deviation within 0.074. The results suggest that interventions exert minimal influence on agent performance in benign tasks, implying that the reduction in BTP reported in Section 5 is primarily attributable to their effect on backdoor attacks. It is worth noting that the lack of BTP improvement under interventions is expected, as their primary objective is to enhance the DRL agent's continual learning capability and alleviate overfitting to specific tasks, which may occasionally lead to a reduction in BTP.

Meanwhile, the performance differences induced by combinations are -0.027, -0.002, -0.001, -0.026, and -0.047, with a maximum standard deviation of 0.192. This suggests that combinations cause a slightly drop in BTP compared to individual interventions, yet the impact remains marginal and does not confound the analysis of their effects on backdoor attacks. We also evaluate the setting where all 7 interventions are combined (*All*) and observe a BTP drop of 0.094 with a standard deviation of 0.268. This indicates that excessive combination of interventions is detrimental to DRL training.

Table 4: Under conventional DRL training, interventions exhibit negligible impact on BTP, with certain combinations causing only slight performance variations.

| Intervention | BTP (mean ± standard deviation) | Difference |
|---|---|---|
| *None* | 0.989 ± 0.032 | +0.000 |
| *Shrink & Perturb* | 0.991 ± 0.025 | +0.002 |
| *Layer Norm* | 0.985 ± 0.074 | -0.004 |
| *Weight Clipping* | 0.982 ± 0.070 | -0.007 |
| *Spectral Norm* | 0.983 ± 0.027 | -0.006 |
| *Weight Decay* | 0.987 ± 0.036 | -0.002 |
| *ReDo* | 0.979 ± 0.055 | -0.012 |
| *SAM* | 0.977 ± 0.068 | -0.010 |
| **Combination** | **BTP** (mean ± standard deviation) | **Difference** |
| *Swiss Cheese* | 0.962 ± 0.192 | -0.027 |
| *Plastic* | 0.987 ± 0.053 | -0.002 |
| *Lac* | 0.988 ± 0.035 | -0.001 |
| *SLac* | 0.963 ± 0.100 | -0.026 |
| *SSW* | 0.942 ± 0.079 | -0.047 |
| *All* | 0.895 ± 0.268 | -0.094 |

## G  DETAILS OF RANKING

**Motivation for Ranking.** We present the effects of interventions on the three pathological characteristics using a ranking-based presentation, motivated by two considerations:

- Task Dimension: The raw metric values exhibit substantial variation across tasks. For example, the range of loss landscape sharpness spans from -1.677 to 2.651 in `Task41`, but from -25837.760 to 36426.301 in `Task38`. The differences in metric scales across tasks may compromise the accuracy of comparisons, since tasks with larger numerical ranges dominate the aggregated analysis.

- Pathology Dimension: The three pathologies have inherently different scales, which complicates cross-metric interpretation. Sorting within each scenario serves as an approximate normalization step, mitigating the influence of scale differences and enabling clearer, more consistent visualization in heatmaps (such as Figure 14).

**Ranking Criteria.** Aligned with prior plasticity studies (Lyle et al., 2023; Sokar et al., 2023; Dohare et al., 2024; Klein et al., 2024), for the pathological characteristics weight magnitude and loss landscape sharpness, smaller values correspond to higher intervention rankings, whereas for effective rank, larger values correspond to higher rankings. The highest rank is 1 and the lowest is 8, meaning that the average ranking ranges from 1 to 8. For example, in Figure 14(a), *Weight Clipping* is ranked #1 for Task0, indicating that in this backdoor attack scenario, compared with other intervention settings, it reduces the weight magnitude to the lowest value.

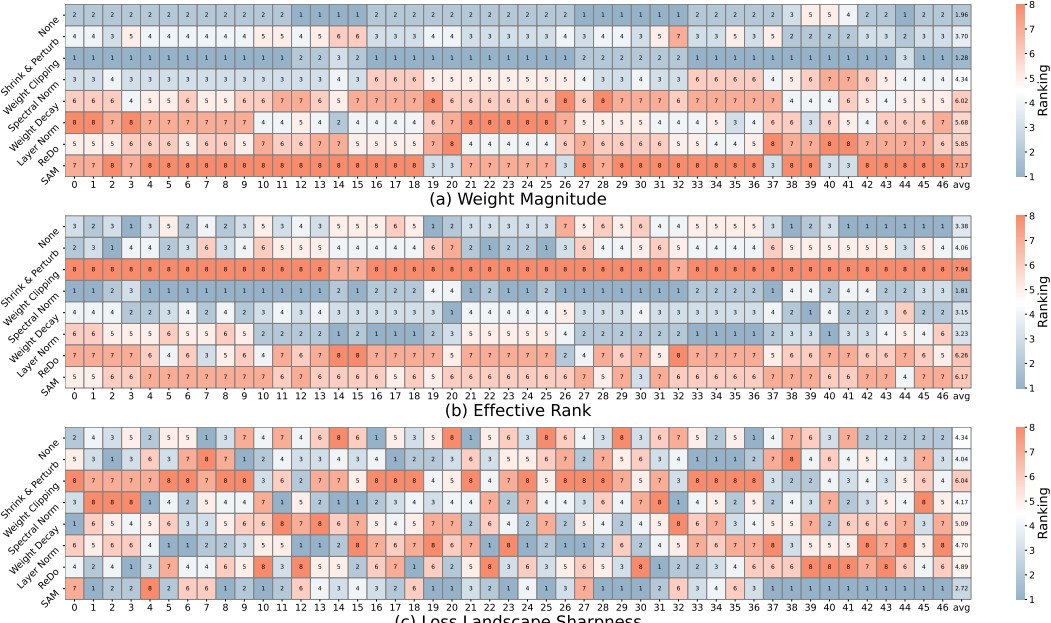

Figure 14: Impact of interventions on backdoor attacks across three pathological characteristics (weight magnitude, effective rank, and loss landscape sharpness). The x-axis corresponds to 47 backdoor task indices, and the y-axis represents 8 intervention settings.

# H NORMALIZED DOT PRODUCT

For a batch of state inputs, we first record the gradient matrix of the parameters with respect to the loss. Then, for each pair of states $s_i$ and $s_j$, we compute the dot product of their gradients:

$$DP[i,j] = \langle \nabla_\theta L(\theta, s_i), \nabla_\theta L(\theta, s_j) \rangle,$$

where $\nabla_\theta L(\theta, s_i)$ denotes the gradient of the loss with respect to the actor network's parameters $\theta$ for state $s_i$. Next, we compute the $\ell_2$ norm of each state's gradient $\|\nabla_\theta L(\theta, s_i)\|_2$. By normalizing the dot products with the corresponding norms, we obtain the normalized gradient dot product matrix (Lyle et al., 2023):

$$DP_{\text{norm}}[i,j] = \frac{\langle \nabla_\theta L(\theta, s_i), \nabla_\theta L(\theta, s_j) \rangle}{\|\nabla_\theta L(\theta, s_i)\|_2 \, \|\nabla_\theta L(\theta, s_j)\|_2}.$$

Finally, we record the mean value of $DP_{\text{norm}}$ across the batch of transition date for analysis.

# I THEORETICAL PROOF

## I.1 PROOF OF THE THEOREM.

**Theorem 1** (*SAM* Amplifies Backdoor Influence in DRL Training): *Let $g_i = \nabla_\theta \ell(\theta; z_i)$ be the gradient for a state $z_i$, $g$ be the mini-batch gradient, and $\bar{H}$ be the average mini-batch Hessian. Let the Empirical Risk Minimization (ERM) and SAM updates be $\Delta\theta_{\mathrm{ERM}} = -\eta g$ and $\Delta\theta_{\mathrm{SAM}} \approx -\eta(g + \frac{\rho}{\|g\|}\bar{H}g)$, respectively. Then, the influence of a backdoor state $z_i$ on the update, when projected onto the backdoor direction $u_b$, satisfies:*

$$\left| u_b^\top \frac{d}{d\varepsilon} \Delta\theta_{\mathrm{SAM}}(0) \right| > \left| u_b^\top \frac{d}{d\varepsilon} \Delta\theta_{\mathrm{ERM}}(0) \right|.$$

*Specifically, SAM influence is amplified by a factor greater than one:*

$$u_b^\top \frac{d}{d\varepsilon} \Delta\theta_{\mathrm{SAM}}(0) = \underbrace{\left( u_b^\top \frac{d}{d\varepsilon} \Delta\theta_{\mathrm{ERM}}(0) \right)}_{-\eta\alpha_i} \cdot \left( 1 + \frac{\rho\lambda_b\|r\|^2}{\|g\|^3} \right).$$

**Assumptions.** The theorem holds under the following assumptions.

- **A1** (Backdoor Gradient Homogeneity): For any backdoor state $z_i$, its gradient $g_i = \alpha_i u_b$ for a fixed unit vector $u_b$ and scalar $\alpha_i > 0$.
- **A2** (Misalignment): The batch gradient can be decomposed as $g = \beta_b u_b + r$, where $u_b^\top r = 0$, $\beta_b > 0$, and the clean residual $\|r\| > 0$.
- **A3** (Curvature Concentration): $u_b$ is an eigenvector of the average Hessian, $\bar{H}u_b = \lambda_b u_b$ with $\lambda_b > 0$, and $u_b^\top \bar{H} r = 0$.

**Proof.** *(1) Upweighting and ERM Influence.* Define the upweighted batch gradient

$$g(\varepsilon) = \frac{1}{B} \sum_{k=1}^{B} \nabla\ell(\theta; z_k) + \varepsilon \nabla\ell(\theta; z_i) = g + \varepsilon g_i.$$

The ERM update is $\Delta\theta_{\mathrm{ERM}}(\varepsilon) = -\eta g(\varepsilon)$, hence

$$\frac{d}{d\varepsilon} \Delta\theta_{\mathrm{ERM}}(\varepsilon)\bigg|_{\varepsilon=0} = -\eta g_i.$$

*(2) SAM Effective Gradient and Its Sensitivity.* For *SAM*, set $v(\varepsilon) := \frac{g(\varepsilon)}{\|g(\varepsilon)\|}$ and $\varepsilon^*(\varepsilon) := \rho v(\varepsilon)$. Using the per-sample first-order Taylor expansion at $\theta$,

$$\nabla\ell(\theta + \varepsilon^*(\varepsilon); z_k) \approx g_k + H_k \varepsilon^*(\varepsilon),$$

and summing over $k$ gives the *SAM*'s effective gradient

$$\tilde{g}(\varepsilon) \approx g(\varepsilon) + \bar{H}\varepsilon^*(\varepsilon) = g(\varepsilon) + \rho\bar{H}v(\varepsilon).$$

Thus, the *SAM* update is $\Delta\theta_{\mathrm{SAM}}(\varepsilon) = -\eta\tilde{g}(\varepsilon)$ and

$$\frac{d}{d\varepsilon} \Delta\theta_{\mathrm{SAM}}(\varepsilon)\bigg|_{\varepsilon=0} = -\eta\left[ \frac{d}{d\varepsilon} g(\varepsilon)\bigg|_0 + \rho\bar{H}\frac{d}{d\varepsilon}v(\varepsilon)\bigg|_0 \right].$$

Since $\frac{d}{d\varepsilon}g(\varepsilon)|_0 = g_i$, it remains to compute $dv/d\varepsilon$ at 0. Recall $v(\varepsilon) = \frac{g+\varepsilon g_i}{\|g+\varepsilon g_i\|}$. Differentiating the normalized vector at $\varepsilon = 0$ yields

$$\frac{d}{d\varepsilon} v(\varepsilon)\bigg|_0 = \frac{1}{\|g\|}\left( I - vv^\top \right) g_i,$$

with $v := g/\|g\|$. Therefore,

$$\frac{d}{d\varepsilon} \Delta\theta_{\mathrm{SAM}}(\varepsilon)\bigg|_0 = -\eta\left[ g_i + \frac{\rho\bar{H}}{\|g\|}\left( g_i - v\left( v^\top g_i \right) \right) \right].$$

*(3) Projection on the Backdoor Direction.* Under the assumptions **A1-A3**, write $g_i = \alpha_i u_b$, $g = \beta_b u_b + r$ with $u_b^\top r = 0$, and $\bar{H}u_b = \lambda_b u_b$, $u_b^\top \bar{H} r = 0$. Then $v = \frac{g}{\|g\|} = \frac{\beta_b}{\|g\|}u_b + \frac{r}{\|g\|}$ and

$$v^\top g_i = \left( \frac{\beta_b}{\|g\|}u_b^\top + \frac{r^\top}{\|g\|} \right) \alpha_i u_b = \alpha_i \frac{\beta_b}{\|g\|}.$$

Furthermore,

$$u_b^\top \bar{H} \left( \frac{g_i}{\|g\|} \right) = \frac{\alpha_i}{\|g\|} u_b^\top \bar{H} u_b = \frac{\alpha_i \lambda_b}{\|g\|},$$

and

$$u_b^\top \bar{H} v = \frac{1}{\|g\|} u_b^\top \bar{H}(\beta_b u_b + r) = \frac{\beta_b \lambda_b}{\|g\|} + \frac{u_b^\top \bar{H} r}{\|g\|} = \frac{\beta_b \lambda_b}{\|g\|}.$$

Therefore,

$$u_b^\top \frac{d}{d\varepsilon}\Delta\theta_{\text{SAM}}(\varepsilon)\bigg|_0 = -\eta\,\alpha_i \left[ 1 + \frac{\rho\lambda_b}{\|g\|} \left( 1 - \frac{\beta_b^2}{\|g\|^2} \right) \right].$$

Since $\|g\|^2 = \beta_b^2 + \|r\|^2$ and $\|r\| > 0$ by (**A2**) with

$$1 + \frac{\rho\lambda_b}{\|g\|} \left( 1 - \frac{\beta_b^2}{\|g\|^2} \right) = 1 + \frac{\rho\lambda_b}{\|g\|^3} \|r\|^2 > 1,$$

the bracket is strictly larger than 1, proving that the magnitude of the *SAM* influence along $u_b$ exceeds the *ERM* influence $-\eta\,\alpha_i$: Therefore, the magnitude of the *SAM* update influence along $u_b$ is strictly larger than that of *ERM*, i.e.,

$$\left| u_b^\top \frac{d}{d\varepsilon}\Delta\theta_{\text{SAM}}(0) \right| > \left| u_b^\top \frac{d}{d\varepsilon}\Delta\theta_{\text{ERM}}(0) \right|.$$

This establishes that, when $g$ is not fully aligned with $u_b$, *SAM* assigns strictly larger effective influence to a backdoor state in the backdoor direction than *ERM* does, hence is more favorable to backdoor injection. ∎

## I.2 RATIONALE FOR THE ASSUMPTIONS

**Rationale for A1.** This assumption models the core mechanism of a backdoor attack. An effective backdoor is typically induced by stamping a consistent trigger onto various samples, which is designed to produce a strong and uniform signal for a target class (Doan et al., 2021). This also holds in the context of DRL backdoor attacks. It is therefore reasonable to posit that the gradients originating from these poisoned samples are closely aligned along a single, dominant "backdoor direction" ($u_b$). The scalars $\{\alpha_i\}$ account for minor variations in gradient magnitude across different samples while preserving the shared directionality.

**Rationale for A2.** This assumption captures the optimization dynamics during the early phases of training, a regime often studied in the context of feature learning (Zhang et al., 2024; Hong et al., 2020; Doan et al., 2021). At this stage, the model has not yet converged to the backdoor feature. The mini-batch gradient ($g$) is thus a composite of two distinct signals: the backdoor component ($\beta_b u_b$) driven by the few backdoor transitions, and a residual component ($r$) driven by the majority of benign transitions. The condition $\|r\| > 0$ is central, as it formally defines this "early stage" where the benign task signal is still present and the overall gradient has not yet fully aligned with the backdoor direction.

**Rationale for A3.** This is a structural assumption on the geometry of the loss landscape, motivated by the observation that backdoor features create sharp, shortcut-like structures Yang et al. (2022); Pu et al. (2024). It is plausible that such a dominant feature direction ($u_b$) would align with a principal eigenvector of the Hessian, corresponding to a direction of high curvature ($\lambda_b$). The orthogonality condition ($u_b^\top \bar{H} r = 0$) serves as a simplifying assumption that decouples the curvature of the backdoor and benign features. Such assumptions about the Hessian's structure are common in theoretical analyses of deep learning to ensure mathematical tractability.

# J SUPPLEMENTARY INVESTIGATION IN MPE

To further investigate the impacts of interventions on DRL backdoor attacks across different algorithms and tasks, we conduct an extended study. Specifically, we conduct evaluations in two multi-agent competitive tasks in Multi Particle Environments (MPE) (Lowe et al., 2017), Predator-Prey and WorldComm, involving four backdoor tasks (see Task47-Task50 in Table 10). In contrast to stochastic algorithms such as PPO, we use deterministic algorithms here, namely DDPG and MADDPG, which correspond to the distributed and centralized training paradigms in multi-agent reinforcement learning. The three intervention settings considered are *None*, *Layer Normalization*, and *SAM*, with the DRL backdoor attack being UNIDOOR.

The results in Table 5 generally align with the main findings presented in Section 5: *Layer Normalization* exhibits a suppressive effect, whereas *SAM* promotes backdoor attacks in *TM-Post*. One exception is that *SAM* shows a suppressing effect in some *TM-Scratch* scenarios, leading to a decrease in BTP. This is because *SAM*'s facilitation of rapid backdoor pathway formation causes the backdoor task to dominate training in *TM-Scratch* (Ma et al., 2025), thereby interfering with the agent's learning of the benign task and, in some cases, leading to training collapse. This further underscores that our findings on *SAM* are primarily relevant to *TM-Post*, and that the *SCC* framework is intended specifically for this scenario.

Table 5: The impacts of three intervention settings (i.e., *None*, *Layer Normalization*, and *SAM* ) on DRL backdoor attacks in MPE. Orange text indicates a significant promoting backdoor threat, Blue text indicates a significant mitigating backdoor threat.

| Algorithm | Threat Model | Intervention | Predator-Prey | | WorldCom | |
|---|---|---|---|---|---|---|
| | | | ASR ↑ | BTP ↑ | ASR ↑ | BTP ↑ |
| DDPG | Conventional Training | *None* | 0.000 | 1.000 | 0.000 | 0.987 |
| | | *Layer Normalization* | 0.000 | 1.000 | 0.000 | 1.000 |
| | | *SAM* | 0.000 | 1.000 | 0.000 | 1.000 |
| | TM-Scratch | *None* | 0.665 | 0.983 | 0.549 | 0.999 |
| | | *Layer Normalization* | 0.212 | 1.000 | 0.185 | 1.000 |
| | | *SAM* | 0.636 | 0.937 | 0.540 | 0.903 |
| | TM-Post | *None* | 0.405 | 1.000 | 0.499 | 0.950 |
| | | *Layer Normalization* | 0.417 | 1.000 | 0.469 | 0.991 |
| | | *SAM* | 0.466 | 0.986 | 0.607 | 0.920 |
| MADDPG | Conventional Training | *None* | 0.000 | 1.000 | 0.000 | 0.981 |
| | | *Layer Normalization* | 0.000 | 0.945 | 0.000 | 0.944 |
| | | *SAM* | 0.000 | 1.000 | 0.000 | 1.000 |
| | TM-Scratch | *None* | 0.654 | 0.958 | 0.497 | 1.000 |
| | | *Layer Normalization* | 0.007 | 0.849 | 0.104 | 1.000 |
| | | *SAM* | 0.642 | 0.874 | 0.501 | 0.976 |
| | TM-Post | *None* | 0.000 | 1.000 | 0.011 | 0.974 |
| | | *Layer Normalization* | 0.000 | 0.773 | 0.014 | 0.856 |
| | | *SAM* | 0.171 | 1.000 | 0.212 | 1.000 |

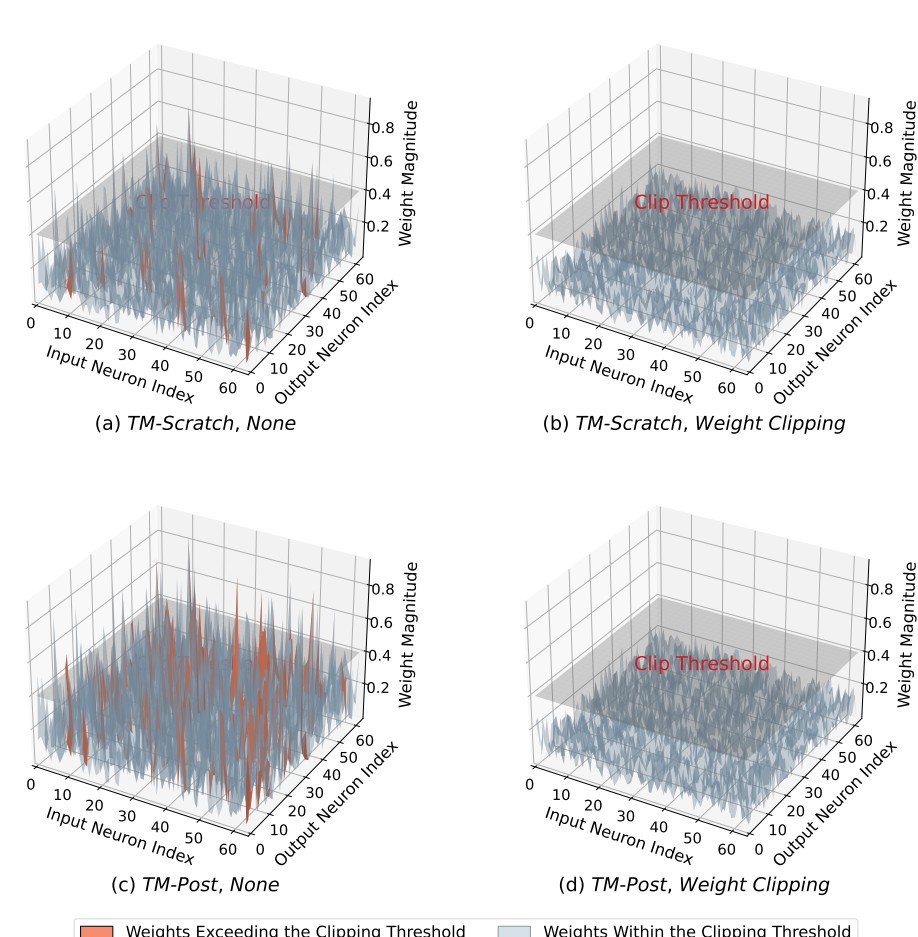

(a) *TM-Scratch, None*      (b) *TM-Scratch, Weight Clipping*

(c) *TM-Post, None*      (d) *TM-Post, Weight Clipping*

■ Weights Exceeding the Clipping Threshold    ■ Weights Within the Clipping Threshold

Figure 15: 3D visualization illustrates the differences in weight magnitude between *TM-Scratch* and *TM-Post* (cf. (a) and (c)). In *TM-Post*, the overall weight magnitude of the actor network are substantially larger than in *TM-Scratch*, causing *Weight Clipping* to clip more weights per iteration, thereby impacting activation pathways for both benign and backdoor tasks. Moreover, the higher weight magnitude reduces the parameter flexibility of the actor network in *TM-Post*, intensifying the competition and conflict when reconstructing activation pathways for both benign and backdoor tasks. Consequently, *Weight Clipping* mitigates backdoor attacks more effectively in *TM-Post*.

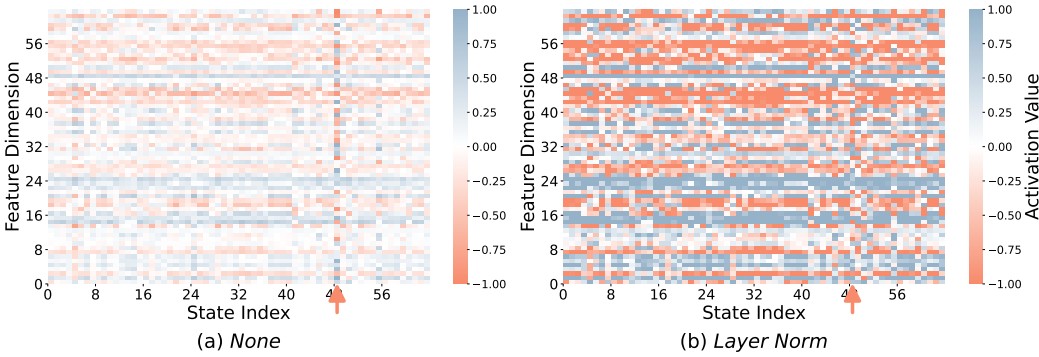

(a) *None*      (b) *Layer Norm*

Figure 16: Without intervention (c.f., (a)), the agent's activation for the backdoor state (red arrow) differs markedly from that of benign states. With *Layer Normalization*, this disparity is substantially reduced (c.f., (b)), thereby lowering the agent's sensitivity to triggers.

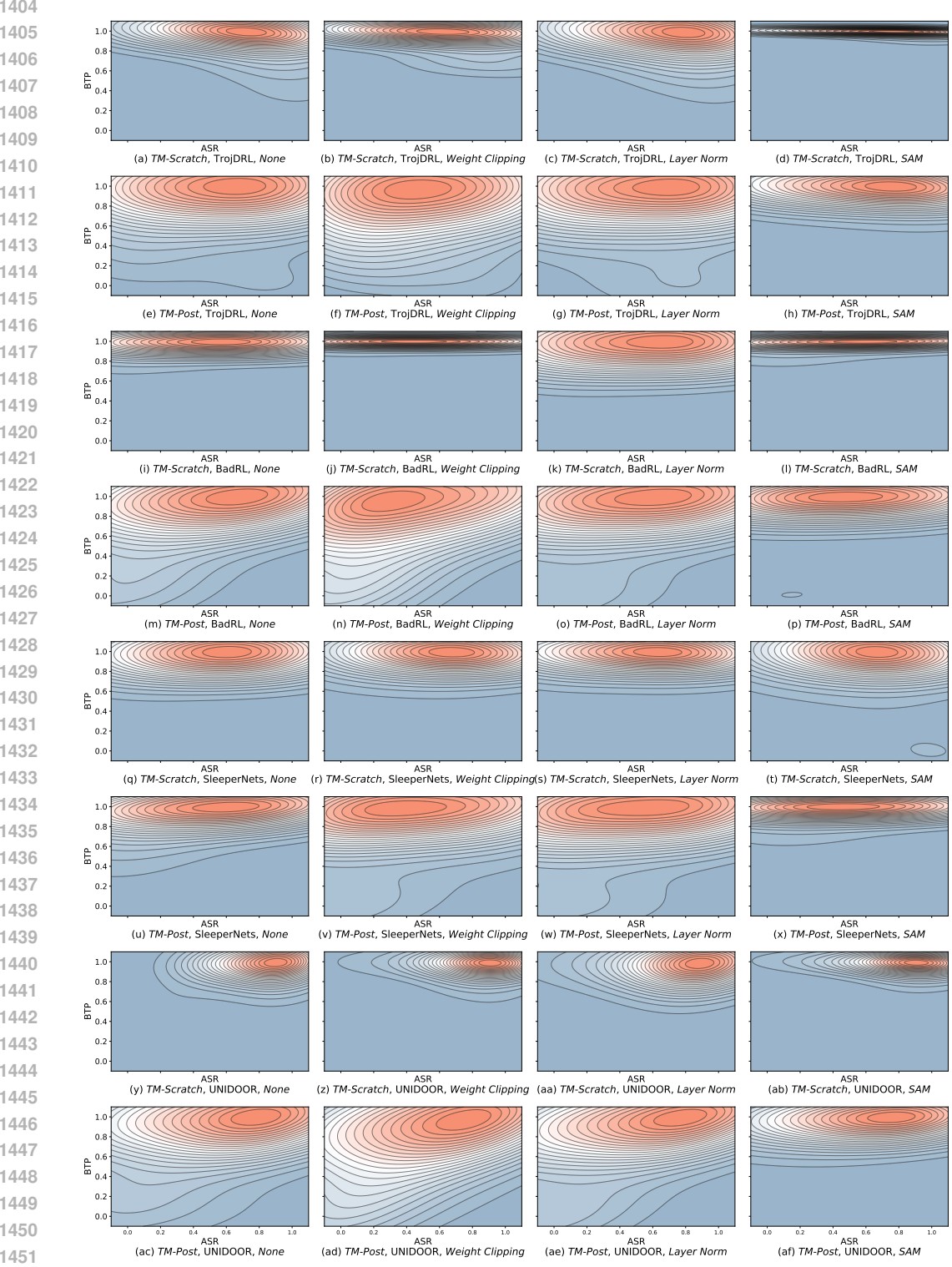

Figure 17: Contour plots of attack performance. Peaks closer to the top-right corner indicate stronger attack performance, characterized by increased ASR and BTP. The four columns in the figure correspond to the effects of four intervention settings (*None*, *Weight Clipping*, *Layer Normalization*, and *SAM*) on two threat models (*TM-Scratch* and *TM-Post*) and three DRL backdoor attack methods (TrojDRL, BadRL, and UNIDOOR).

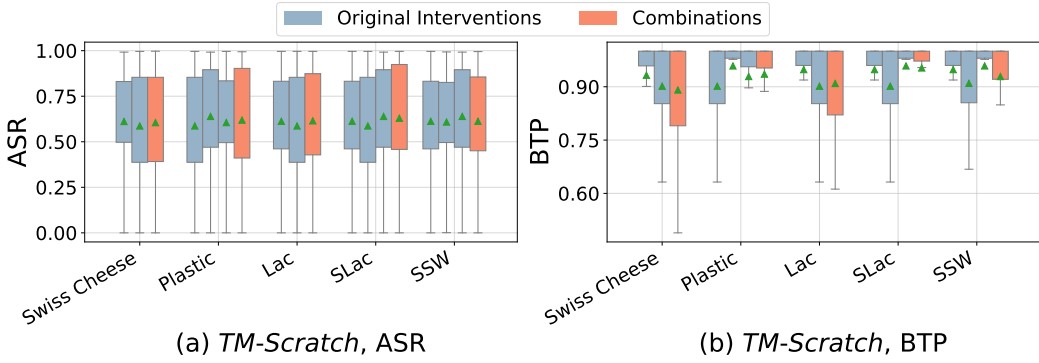

(a) *TM-Scratch*, ASR

(b) *TM-Scratch*, BTP

Figure 18: Impact of combinations in *TM-Scratch*.

Table 6: Impact of interventions in *TM-Scratch*. Orange text indicates a significant promoting backdoor threat, Blue text indicates a significant mitigating backdoor threat.

| Intervention | Method | Classic Control Tasks | | Physics Control Tasks | | Robotic Tasks | |
|---|---|---|---|---|---|---|---|
| | | ASR ↑ | BTP ↑ | ASR ↑ | BTP ↑ | ASR ↑ | BTP ↑ |
| *None* | TrojDRL | 0.673 ±0.138 | 0.997 ±0.003 | 0.543 ±0.121 | 0.972 ±0.045 | 0.974 ±0.020 | 0.720 ±0.181 |
| | BadRL | 0.613 ±0.183 | 0.956 ±0.109 | 0.435 ±0.424 | 0.988 ±0.015 | 0.025 ±0.039 | 0.992 ±0.012 |
| | SleeperNets | 0.705 ±0.186 | 0.935 ±0.167 | 0.620 ±0.105 | 0.991 ±0.009 | 0.493 ±0.140 | 0.860 ±0.172 |
| | UNIDOOR | 0.801 ±0.096 | 0.996 ±0.007 | 0.834 ±0.108 | 0.908 ±0.139 | 0.888 ±0.113 | 0.897 ±0.074 |
| | Average | 0.698 ±0.151 | 0.971 ±0.072 | 0.608 ±0.190 | 0.965 ±0.052 | 0.595 ±0.078 | 0.867 ±0.110 |
| *Shrink & Perturb* | TrojDRL | 0.696 ±0.161 | 0.996 ±0.005 | 0.502 ±0.148 | 0.988 ±0.015 | 0.868 ±0.105 | 0.695 ±0.203 |
| | BadRL | 0.591 ±0.200 | 0.999 ±0.002 | 0.433 ±0.429 | 0.990 ±0.010 | 0.008 ±0.014 | 0.949 ±0.079 |
| | SleeperNets | 0.700 ±0.184 | 0.934 ±0.167 | 0.557 ±0.136 | 0.992 ±0.008 | 0.290 ±0.120 | 0.901 ±0.067 |
| | UNIDOOR | 0.797 ±0.129 | 0.993 ±0.013 | 0.812 ±0.012 | 0.987 ±0.016 | 0.751 ±0.159 | 0.790 ±0.199 |
| | Average | 0.696 ±0.169 | 0.981 ±0.047 | 0.576 ±0.181 | 0.989 ±0.012 | 0.479 ±0.099 | 0.834 ±0.137 |
| *Weight Clipping* | TrojDRL | 0.616 ±0.160 | 0.995 ±0.010 | 0.507 ±0.084 | 0.994 ±0.006 | 0.883 ±0.118 | 0.824 ±0.103 |
| | BadRL | 0.571 ±0.193 | 0.995 ±0.010 | 0.425 ±0.425 | 0.990 ±0.009 | 0.004 ±0.004 | 0.955 ±0.064 |
| | SleeperNets | 0.724 ±0.175 | 0.898 ±0.165 | 0.612 ±0.099 | 0.997 ±0.004 | 0.407 ±0.078 | 0.956 ±0.053 |
| | UNIDOOR | 0.818 ±0.107 | 0.983 ±0.027 | 0.805 ±0.039 | 0.955 ±0.022 | 0.849 ±0.056 | 0.861 ±0.179 |
| | Average | 0.682 ±0.159 | 0.967 ±0.053 | 0.587 ±0.162 | 0.984 ±0.010 | 0.536 ±0.064 | 0.899 ±0.100 |
| *Spectral Normalization* | TrojDRL | 0.654 ±0.148 | 0.958 ±0.055 | 0.523 ±0.206 | 0.962 ±0.012 | 0.870 ±0.127 | 0.790 ±0.180 |
| | BadRL | 0.568 ±0.162 | 0.962 ±0.050 | 0.421 ±0.421 | 0.961 ±0.040 | 0.012 ±0.018 | 0.922 ±0.083 |
| | SleeperNets | 0.694 ±0.186 | 0.911 ±0.179 | 0.493 ±0.233 | 0.986 ±0.016 | 0.584 ±0.140 | 0.771 ±0.110 |
| | UNIDOOR | 0.787 ±0.102 | 0.973 ±0.035 | 0.695 ±0.133 | 0.939 ±0.076 | 0.812 ±0.116 | 0.796 ±0.131 |
| | Average | 0.676 ±0.149 | 0.951 ±0.080 | 0.533 ±0.248 | 0.962 ±0.036 | 0.569 ±0.100 | 0.820 ±0.131 |
| *Weight Decay* | TrojDRL | 0.668 ±0.148 | 0.997 ±0.004 | 0.543 ±0.121 | 0.972 ±0.045 | 0.872 ±0.092 | 0.727 ±0.162 |
| | BadRL | 0.604 ±0.196 | 0.997 ±0.004 | 0.435 ±0.424 | 0.988 ±0.015 | 0.014 ±0.011 | 0.955 ±0.078 |
| | SleeperNets | 0.705 ±0.186 | 0.934 ±0.166 | 0.620 ±0.105 | 0.991 ±0.009 | 0.395 ±0.124 | 0.865 ±0.090 |
| | UNIDOOR | 0.804 ±0.140 | 0.993 ±0.015 | 0.827 ±0.109 | 0.988 ±0.017 | 0.728 ±0.207 | 0.789 ±0.213 |
| | Average | 0.695 ±0.168 | 0.980 ±0.047 | 0.606 ±0.190 | 0.985 ±0.021 | 0.502 ±0.109 | 0.834 ±0.136 |
| *Layer Normalization* | TrojDRL | 0.738 ±0.192 | 0.954 ±0.110 | 0.435 ±0.135 | 0.999 ±0.002 | 0.879 ±0.098 | 0.754 ±0.131 |
| | BadRL | 0.661 ±0.235 | 0.790 ±0.287 | 0.431 ±0.430 | 0.996 ±0.006 | 0.011 ±0.015 | 0.879 ±0.146 |
| | SleeperNets | 0.727 ±0.188 | 0.932 ±0.169 | 0.496 ±0.114 | 1.000 ±0.000 | 0.107 ±0.081 | 0.914 ±0.095 |
| | UNIDOOR | 0.756 ±0.147 | 0.911 ±0.152 | 0.708 ±0.145 | 0.988 ±0.020 | 0.823 ±0.082 | 0.836 ±0.111 |
| | Average | 0.720 ±0.191 | 0.897 ±0.179 | 0.517 ±0.206 | 0.996 ±0.007 | 0.455 ±0.069 | 0.846 ±0.121 |
| *ReDo* | TrojDRL | 0.683 ±0.149 | 0.986 ±0.022 | 0.517 ±0.153 | 0.996 ±0.004 | 0.859 ±0.119 | 0.710 ±0.167 |
| | BadRL | 0.589 ±0.191 | 0.993 ±0.016 | 0.438 ±0.427 | 0.989 ±0.012 | 0.008 ±0.013 | 0.956 ±0.076 |
| | SleeperNets | 0.699 ±0.190 | 0.932 ±0.167 | 0.598 ±0.101 | 0.971 ±0.043 | 0.404 ±0.122 | 0.874 ±0.148 |
| | UNIDOOR | 0.787 ±0.127 | 0.989 ±0.021 | 0.827 ±0.109 | 0.990 ±0.012 | 0.728 ±0.195 | 0.776 ±0.233 |
| | Average | 0.689 ±0.164 | 0.975 ±0.056 | 0.595 ±0.198 | 0.987 ±0.018 | 0.500 ±0.112 | 0.829 ±0.156 |
| *SAM* | TrojDRL | 0.679 ±0.183 | 0.995 ±0.006 | 0.545 ±0.171 | 0.998 ±0.002 | 0.951 ±0.034 | 0.963 ±0.033 |
| | BadRL | 0.611 ±0.205 | 0.998 ±0.003 | 0.433 ±0.433 | 0.990 ±0.010 | 0.041 ±0.041 | 0.929 ±0.072 |
| | SleeperNets | 0.715 ±0.177 | 0.934 ±0.168 | 0.514 ±0.206 | 0.996 ±0.005 | 0.474 ±0.178 | 0.846 ±0.155 |
| | UNIDOOR | 0.784 ±0.134 | 0.982 ±0.039 | 0.825 ±0.115 | 0.983 ±0.024 | 0.941 ±0.030 | 0.909 ±0.054 |
| | Average | 0.697 ±0.175 | 0.977 ±0.054 | 0.579 ±0.231 | 0.992 ±0.010 | 0.602 ±0.071 | 0.912 ±0.078 |

Table 7: Impact of interventions in *TM-Post*. Orange text indicates a significant promoting backdoor threat, Blue text indicates a significant mitigating backdoor threat.

| Intervention | Method | Classic Control Tasks | | Physics Control Tasks | | Robotic Tasks | |
|---|---|---|---|---|---|---|---|
| | | ASR ↑ | BTP ↑ | ASR ↑ | BTP ↑ | ASR ↑ | BTP ↑ |
| *None* | TrojDRL | 0.749 ±0.186 | 1.000 ±0.000 | 0.184 ±0.101 | 1.000 ±0.000 | 0.344 ±0.311 | 0.722 ±0.280 |
| | BadRL | 0.698 ±0.196 | 1.000 ±0.000 | 0.277 ±0.271 | 1.000 ±0.000 | 0.005 ±0.006 | 0.833 ±0.217 |
| | SleeperNets | 0.724 ±0.188 | 1.000 ±0.000 | 0.109 ±0.143 | 1.000 ±0.000 | 0.099 ±0.102 | 0.707 ±0.164 |
| | UNIDOOR | 0.772 ±0.171 | 1.000 ±0.000 | 0.326 ±0.177 | 1.000 ±0.000 | 0.264 ±0.210 | 0.716 ±0.258 |
| | Average | 0.736 ±0.185 | 1.000 ±0.000 | 0.234 ±0.173 | 1.000 ±0.000 | 0.178 ±0.157 | 0.745 ±0.230 |
| *Shrink & Perturb* | TrojDRL | 0.731 ±0.198 | 1.000 ±0.000 | 0.211 ±0.046 | 1.000 ±0.000 | 0.422 ±0.317 | 0.462 ±0.373 |
| | BadRL | 0.681 ±0.209 | 1.000 ±0.000 | 0.202 ±0.211 | 1.000 ±0.000 | 0.000 ±0.000 | 0.547 ±0.387 |
| | SleeperNets | 0.739 ±0.179 | 1.000 ±0.000 | 0.130 ±0.121 | 1.000 ±0.000 | 0.001 ±0.001 | 0.580 ±0.313 |
| | UNIDOOR | 0.753 ±0.192 | 1.000 ±0.000 | 0.239 ±0.168 | 1.000 ±0.000 | 0.009 ±0.017 | 0.474 ±0.360 |
| | Average | 0.726 ±0.195 | 1.000 ±0.000 | 0.195 ±0.137 | 1.000 ±0.000 | 0.108 ±0.084 | 0.516 ±0.358 |
| *Weight Clipping* | TrojDRL | 0.585 ±0.287 | 0.855 ±0.149 | 0.311 ±0.024 | 0.832 ±0.169 | 0.344 ±0.356 | 0.510 ±0.382 |
| | BadRL | 0.464 ±0.320 | 0.812 ±0.224 | 0.292 ±0.272 | 0.838 ±0.163 | 0.002 ±0.003 | 0.577 ±0.405 |
| | SleeperNets | 0.715 ±0.171 | 0.988 ±0.032 | 0.062 ±0.073 | 1.000 ±0.000 | 0.002 ±0.003 | 0.566 ±0.399 |
| | UNIDOOR | 0.653 ±0.209 | 0.831 ±0.183 | 0.439 ±0.087 | 0.824 ±0.176 | 0.034 ±0.048 | 0.522 ±0.384 |
| | Average | 0.604 ±0.247 | 0.871 ±0.147 | 0.276 ±0.114 | 0.873 ±0.127 | 0.096 ±0.102 | 0.544 ±0.392 |
| *Spectral Normalization* | TrojDRL | 0.615 ±0.216 | 0.993 ±0.011 | 0.195 ±0.195 | 0.999 ±0.001 | 0.490 ±0.354 | 0.539 ±0.385 |
| | BadRL | 0.584 ±0.244 | 0.993 ±0.011 | 0.273 ±0.273 | 0.999 ±0.001 | 0.006 ±0.009 | 0.593 ±0.382 |
| | SleeperNets | 0.664 ±0.180 | 0.981 ±0.039 | 0.219 ±0.220 | 0.998 ±0.002 | 0.011 ±0.017 | 0.576 ±0.302 |
| | UNIDOOR | 0.659 ±0.226 | 0.985 ±0.023 | 0.285 ±0.283 | 0.996 ±0.006 | 0.121 ±0.168 | 0.553 ±0.398 |
| | Average | 0.631 ±0.216 | 0.988 ±0.021 | 0.243 ±0.243 | 0.998 ±0.003 | 0.157 ±0.137 | 0.565 ±0.367 |
| *Weight Decay* | TrojDRL | 0.736 ±0.203 | 1.000 ±0.000 | 0.184 ±0.101 | 1.000 ±0.000 | 0.395 ±0.308 | 0.549 ±0.351 |
| | BadRL | 0.686 ±0.210 | 1.000 ±0.000 | 0.277 ±0.271 | 1.000 ±0.000 | 0.005 ±0.007 | 0.680 ±0.313 |
| | SleeperNets | 0.723 ±0.188 | 1.000 ±0.000 | 0.149 ±0.143 | 1.000 ±0.000 | 0.006 ±0.011 | 0.579 ±0.325 |
| | UNIDOOR | 0.737 ±0.200 | 1.000 ±0.000 | 0.291 ±0.272 | 1.000 ±0.000 | 0.090 ±0.120 | 0.603 ±0.299 |
| | Average | 0.720 ±0.204 | 1.000 ±0.000 | 0.251 ±0.215 | 1.000 ±0.000 | 0.163 ±0.145 | 0.611 ±0.321 |
| *Layer Normalization* | TrojDRL | 0.763 ±0.203 | 0.896 ±0.185 | 0.147 ±0.093 | 1.000 ±0.000 | 0.396 ±0.253 | 0.547 ±0.368 |
| | BadRL | 0.689 ±0.202 | 0.917 ±0.144 | 0.252 ±0.253 | 1.000 ±0.000 | 0.008 ±0.009 | 0.576 ±0.402 |
| | SleeperNets | 0.755 ±0.182 | 0.917 ±0.144 | 0.082 ±0.095 | 1.000 ±0.000 | 0.005 ±0.006 | 0.624 ±0.363 |
| | UNIDOOR | 0.710 ±0.212 | 0.917 ±0.144 | 0.233 ±0.237 | 1.000 ±0.000 | 0.090 ±0.114 | 0.522 ±0.374 |
| | Average | 0.729 ±0.200 | 0.912 ±0.154 | 0.178 ±0.170 | 1.000 ±0.000 | 0.125 ±0.096 | 0.568 ±0.377 |
| *ReDo* | TrojDRL | 0.743 ±0.203 | 0.938 ±0.116 | 0.188 ±0.107 | 1.000 ±0.000 | 0.454 ±0.349 | 0.547 ±0.339 |
| | BadRL | 0.695 ±0.209 | 1.000 ±0.001 | 0.271 ±0.265 | 1.000 ±0.000 | 0.005 ±0.007 | 0.656 ±0.284 |
| | SleeperNets | 0.728 ±0.186 | 1.000 ±0.000 | 0.119 ±0.113 | 1.000 ±0.000 | 0.133 ±0.117 | 0.641 ±0.247 |
| | UNIDOOR | 0.778 ±0.195 | 0.999 ±0.002 | 0.273 ±0.253 | 1.000 ±0.000 | 0.017 ±0.027 | 0.578 ±0.364 |
| | Average | 0.736 ±0.198 | 0.984 ±0.030 | 0.213 ±0.185 | 1.000 ±0.000 | 0.152 ±0.125 | 0.605 ±0.309 |
| *SAM* | TrojDRL | 0.682 ±0.217 | 1.000 ±0.001 | 0.305 ±0.020 | 1.000 ±0.000 | 0.778 ±0.235 | 0.804 ±0.233 |
| | BadRL | 0.641 ±0.215 | 1.000 ±0.001 | 0.270 ±0.276 | 1.000 ±0.000 | 0.058 ±0.089 | 0.803 ±0.283 |
| | SleeperNets | 0.698 ±0.194 | 1.000 ±0.000 | 0.166 ±0.107 | 1.000 ±0.000 | 0.082 ±0.074 | 0.903 ±0.115 |
| | UNIDOOR | 0.715 ±0.193 | 1.000 ±0.000 | 0.438 ±0.074 | 1.000 ±0.000 | 0.384 ±0.124 | 0.745 ±0.283 |
| | Average | 0.684 ±0.205 | 1.000 ±0.000 | 0.295 ±0.119 | 1.000 ±0.000 | 0.326 ±0.131 | 0.814 ±0.229 |

Table 8: Impact of combinations in *TM-Scratch*. Orange text indicates a significant promoting backdoor threat, Blue text indicates a significant mitigating backdoor threat.

| Combination | Method | Classic Control Tasks | | Physics Control Tasks | | Robotic Tasks | |
|---|---|---|---|---|---|---|---|
| | | ASR ↑ | BTP ↑ | ASR ↑ | BTP ↑ | ASR ↑ | BTP ↑ |
| *None* | TrojDRL | 0.673 ±0.138 | 0.997 ±0.003 | 0.543 ±0.121 | 0.972 ±0.045 | 0.974 ±0.020 | 0.720 ±0.181 |
| | BadRL | 0.613 ±0.183 | 0.956 ±0.109 | 0.435 ±0.424 | 0.988 ±0.015 | 0.025 ±0.039 | 0.992 ±0.012 |
| | SleeperNets | 0.705 ±0.186 | 0.935 ±0.167 | 0.460 ±0.105 | 0.991 ±0.009 | 0.493 ±0.140 | 0.860 ±0.172 |
| | UNIDOOR | 0.801 ±0.096 | 0.996 ±0.007 | 0.834 ±0.108 | 0.908 ±0.139 | 0.888 ±0.113 | 0.897 ±0.074 |
| | Average | 0.698 ±0.151 | 0.971 ±0.072 | 0.568 ±0.190 | 0.965 ±0.052 | 0.595 ±0.078 | 0.867 ±0.110 |
| *Swiss Cheese* | TrojDRL | 0.742 ±0.192 | 0.955 ±0.110 | 0.435 ±0.135 | 0.999 ±0.002 | 0.879 ±0.098 | 0.754 ±0.131 |
| | BadRL | 0.664 ±0.235 | 0.791 ±0.286 | 0.431 ±0.430 | 0.996 ±0.006 | 0.011 ±0.015 | 0.879 ±0.146 |
| | SleeperNets | 0.725 ±0.189 | 0.933 ±0.168 | 0.496 ±0.114 | 1.000 ±0.000 | 0.310 ±0.194 | 0.783 ±0.145 |
| | UNIDOOR | 0.763 ±0.147 | 0.911 ±0.152 | 0.708 ±0.145 | 0.988 ±0.020 | 0.823 ±0.082 | 0.836 ±0.111 |
| | Average | 0.723 ±0.191 | 0.897 ±0.179 | 0.517 ±0.206 | 0.996 ±0.007 | 0.506 ±0.097 | 0.813 ±0.133 |
| *Plastic* | TrojDRL | 0.704 ±0.205 | 0.958 ±0.110 | 0.457 ±0.053 | 0.999 ±0.001 | 0.947 ±0.032 | 0.905 ±0.063 |
| | BadRL | 0.612 ±0.233 | 0.921 ±0.145 | 0.442 ±0.442 | 0.906 ±0.139 | 0.046 ±0.053 | 0.932 ±0.117 |
| | SleeperNets | 0.743 ±0.171 | 0.903 ±0.221 | 0.508 ±0.106 | 0.998 ±0.003 | 0.383 ±0.158 | 0.942 ±0.129 |
| | UNIDOOR | 0.734 ±0.188 | 0.952 ±0.112 | 0.821 ±0.015 | 0.992 ±0.014 | 0.942 ±0.029 | 0.873 ±0.110 |
| | Average | 0.698 ±0.209 | 0.933 ±0.147 | 0.557 ±0.154 | 0.974 ±0.039 | 0.580 ±0.068 | 0.913 ±0.105 |
| *Lac* | TrojDRL | 0.720 ±0.185 | 0.957 ±0.111 | 0.544 ±0.195 | 1.000 ±0.000 | 0.872 ±0.092 | 0.727 ±0.162 |
| | BadRL | 0.656 ±0.237 | 0.957 ±0.111 | 0.425 ±0.426 | 0.998 ±0.003 | 0.014 ±0.011 | 0.955 ±0.078 |
| | SleeperNets | 0.756 ±0.152 | 0.892 ±0.168 | 0.556 ±0.100 | 1.000 ±0.000 | 0.268 ±0.122 | 0.825 ±0.089 |
| | UNIDOOR | 0.797 ±0.167 | 0.903 ±0.116 | 0.895 ±0.097 | 0.991 ±0.014 | 0.705 ±0.172 | 0.800 ±0.208 |
| | Average | 0.732 ±0.185 | 0.927 ±0.126 | 0.605 ±0.205 | 0.997 ±0.004 | 0.465 ±0.099 | 0.827 ±0.134 |
| *SLac* | TrojDRL | 0.705 ±0.199 | 0.958 ±0.110 | 0.409 ±0.146 | 0.999 ±0.002 | 0.949 ±0.043 | 0.973 ±0.024 |
| | BadRL | 0.628 ±0.216 | 0.958 ±0.110 | 0.410 ±0.411 | 0.997 ±0.005 | 0.031 ±0.035 | 0.934 ±0.101 |
| | SleeperNets | 0.732 ±0.172 | 0.925 ±0.168 | 0.569 ±0.080 | 0.997 ±0.005 | 0.518 ±0.194 | 0.954 ±0.072 |
| | UNIDOOR | 0.795 ±0.159 | 0.897 ±0.145 | 0.884 ±0.113 | 0.976 ±0.028 | 0.848 ±0.183 | 0.944 ±0.058 |
| | Average | 0.715 ±0.186 | 0.935 ±0.133 | 0.568 ±0.187 | 0.992 ±0.010 | 0.586 ±0.114 | 0.951 ±0.064 |
| *SSW* | TrojDRL | 0.586 ±0.161 | 0.963 ±0.070 | 0.358 ±0.078 | 0.898 ±0.133 | 0.925 ±0.068 | 0.962 ±0.026 |
| | BadRL | 0.559 ±0.183 | 0.962 ±0.070 | 0.404 ±0.403 | 0.864 ±0.168 | 0.046 ±0.047 | 0.939 ±0.088 |
| | SleeperNets | 0.725 ±0.161 | 0.900 ±0.166 | 0.551 ±0.151 | 0.961 ±0.050 | 0.565 ±0.118 | 0.928 ±0.112 |
| | UNIDOOR | 0.797 ±0.164 | 0.981 ±0.022 | 0.743 ±0.116 | 0.800 ±0.128 | 0.889 ±0.130 | 0.893 ±0.112 |
| | Average | 0.667 ±0.167 | 0.952 ±0.082 | 0.514 ±0.187 | 0.881 ±0.120 | 0.606 ±0.091 | 0.930 ±0.084 |

Table 9: Impact of combinations in *TM-Post*.

| Combination | Method | Classic Control Tasks | | Physics Control Tasks | | Robotic Tasks | |
|---|---|---|---|---|---|---|---|
| | | ASR ↑ | BTP ↑ | ASR ↑ | BTP ↑ | ASR ↑ | BTP ↑ |
| *None* | TrojDRL | 0.749 ±0.186 | 1.000 ±0.000 | 0.184 ±0.101 | 1.000 ±0.000 | 0.344 ±0.311 | 0.722 ±0.280 |
| | BadRL | 0.698 ±0.196 | 1.000 ±0.000 | 0.277 ±0.271 | 1.000 ±0.000 | 0.005 ±0.006 | 0.833 ±0.217 |
| | SleeperNets | 0.724 ±0.188 | 1.000 ±0.000 | 0.149 ±0.143 | 1.000 ±0.000 | 0.099 ±0.102 | 0.707 ±0.164 |
| | UNIDOOR | 0.772 ±0.171 | 1.000 ±0.000 | 0.326 ±0.177 | 1.000 ±0.000 | 0.264 ±0.210 | 0.716 ±0.258 |
| | Average | 0.736 ±0.185 | 1.000 ±0.000 | 0.234 ±0.173 | 1.000 ±0.000 | 0.178 ±0.157 | 0.745 ±0.230 |
| *Swiss Cheese* | TrojDRL | 0.768 ±0.204 | 0.913 ±0.150 | 0.147 ±0.093 | 1.000 ±0.000 | 0.396 ±0.253 | 0.547 ±0.368 |
| | BadRL | 0.690 ±0.201 | 0.917 ±0.144 | 0.252 ±0.253 | 1.000 ±0.000 | 0.008 ±0.009 | 0.576 ±0.402 |
| | SleeperNets | 0.750 ±0.185 | 0.917 ±0.144 | 0.082 ±0.095 | 1.000 ±0.000 | 0.005 ±0.006 | 0.624 ±0.363 |
| | UNIDOOR | 0.750 ±0.178 | 0.917 ±0.144 | 0.233 ±0.237 | 1.000 ±0.000 | 0.090 ±0.114 | 0.522 ±0.374 |
| | Average | 0.739 ±0.192 | 0.916 ±0.146 | 0.178 ±0.170 | 1.000 ±0.000 | 0.125 ±0.096 | 0.568 ±0.377 |
| *Plastic* | TrojDRL | 0.721 ±0.217 | 0.934 ±0.123 | 0.128 ±0.129 | 1.000 ±0.000 | 0.882 ±0.120 | 0.716 ±0.405 |
| | BadRL | 0.656 ±0.210 | 0.896 ±0.220 | 0.277 ±0.277 | 1.000 ±0.000 | 0.047 ±0.055 | 0.716 ±0.408 |
| | SleeperNets | 0.741 ±0.183 | 0.979 ±0.056 | 0.119 ±0.119 | 1.000 ±0.000 | 0.178 ±0.140 | 0.779 ±0.201 |
| | UNIDOOR | 0.742 ±0.206 | 0.958 ±0.111 | 0.392 ±0.156 | 1.000 ±0.000 | 0.362 ±0.262 | 0.683 ±0.434 |
| | Average | 0.715 ±0.204 | 0.942 ±0.127 | 0.229 ±0.170 | 1.000 ±0.000 | 0.368 ±0.144 | 0.724 ±0.362 |
| *Lac* | TrojDRL | 0.738 ±0.193 | 0.979 ±0.055 | 0.077 ±0.078 | 1.000 ±0.000 | 0.464 ±0.357 | 0.610 ±0.278 |
| | BadRL | 0.637 ±0.201 | 0.979 ±0.055 | 0.241 ±0.252 | 1.000 ±0.000 | 0.007 ±0.008 | 0.773 ±0.158 |
| | SleeperNets | 0.744 ±0.159 | 1.000 ±0.000 | 0.068 ±0.073 | 1.000 ±0.000 | 0.013 ±0.012 | 0.650 ±0.191 |
| | UNIDOOR | 0.747 ±0.153 | 1.000 ±0.000 | 0.211 ±0.211 | 1.000 ±0.000 | 0.177 ±0.181 | 0.685 ±0.201 |
| | Average | 0.716 ±0.176 | 0.989 ±0.028 | 0.149 ±0.153 | 1.000 ±0.000 | 0.165 ±0.139 | 0.680 ±0.207 |
| *SLac* | TrojDRL | 0.766 ±0.194 | 0.937 ±0.116 | 0.144 ±0.087 | 1.000 ±0.000 | 0.862 ±0.105 | 0.799 ±0.303 |
| | BadRL | 0.692 ±0.204 | 0.936 ±0.117 | 0.276 ±0.278 | 1.000 ±0.000 | 0.044 ±0.050 | 0.790 ±0.311 |
| | SleeperNets | 0.756 ±0.171 | 0.937 ±0.116 | 0.113 ±0.117 | 1.000 ±0.000 | 0.200 ±0.176 | 0.899 ±0.166 |
| | UNIDOOR | 0.745 ±0.192 | 0.916 ±0.145 | 0.294 ±0.223 | 1.000 ±0.000 | 0.564 ±0.253 | 0.776 ±0.324 |
| | Average | 0.740 ±0.190 | 0.932 ±0.124 | 0.207 ±0.176 | 1.000 ±0.000 | 0.417 ±0.146 | 0.816 ±0.276 |
| *SSW* | TrojDRL | 0.677 ±0.177 | 0.999 ±0.002 | 0.142 ±0.149 | 0.976 ±0.028 | 0.916 ±0.041 | 0.914 ±0.161 |
| | BadRL | 0.641 ±0.195 | 0.999 ±0.002 | 0.255 ±0.265 | 0.947 ±0.055 | 0.049 ±0.046 | 0.889 ±0.177 |
| | SleeperNets | 0.707 ±0.166 | 0.991 ±0.020 | 0.213 ±0.220 | 0.984 ±0.017 | 0.089 ±0.069 | 0.961 ±0.056 |
| | UNIDOOR | 0.750 ±0.167 | 0.988 ±0.025 | 0.337 ±0.215 | 0.904 ±0.143 | 0.620 ±0.213 | 0.870 ±0.184 |
| | Average | 0.694 ±0.176 | 0.994 ±0.012 | 0.237 ±0.212 | 0.953 ±0.061 | 0.418 ±0.092 | 0.915 ±0.131 |

Table 10: Design details of the backdoor tasks. `Task0-Task46` correspond to the 47 backdoor tasks in the main investigation, while `Task47-Task50` correspond to the 4 backdoor tasks in the extended investigation.

| Index | Environment | Backdoor Task |
|-------|-------------|---------------|
| Task0 | CartPole | ( $s^{(0)}$, -4.8, push cart to the right ). |
| Task1 | CartPole | ( $s^{(1)}$, 100, push cart to the right ). |
| Task2 | CartPole | ( $s^{(2)}$, -0.42, push cart to the left ). |
| Task3 | CartPole | ( $s^{(3)}$, -100, push cart to the left ). |
| Task4 | Acrobot | ( $s^{(0)}$, -1, apply -1 torque ). |
| Task5 | Acrobot | ( $s^{(1)}$, -1, apply 0 torque ). |
| Task6 | Acrobot | ( $s^{(2)}$, -1, apply 1 torque ). |
| Task7 | Acrobot | ( $s^{(3)}$, -1, apply -1 torque ). |
| Task8 | Acrobot | ( $s^{(4)}$, 12.57, apply 0 torque ). |
| Task9 | Acrobot | ( $s^{(5)}$, 28.27, apply 1 torque ). |
| Task10 | MountainCar | ( $s^{(0)}$, -0.07, not accelerate ). |
| Task11 | MountainCar | ( $s^{(1)}$, 0.07, accelerate to the right ). |
| Task12 | Pendulum | ( $s^{(2)}$, 8, maximum left torque ). |
| Task13 | Pendulum | ( $s^{(1)}$, -1, maximum right torque ). |
| Task14 | Pendulum | ( $s^{(2)}$, -8, maximum right torque ). |
| Task15 | CartPole | ( $s^{(0)}$, -4.8, push cart to the right ), ( $s^{(2)}$, -0.42, push cart to the left ). |
| Task16 | CartPole | ( $s^{(1)}$, 100, push cart to the right ), ( $s^{(3)}$, -100, push cart to the left ). |
| Task17 | CartPole | ( $s^{(0)}$, -4.8, push cart to the right ), ( $s^{(3)}$, -100, push cart to the left ). |
| Task18 | CartPole | ( $s^{(1)}$, 100, push cart to the right ), ( $s^{(2)}$, -0.42, push cart to the left ). |
| Task19 | CartPole | ( $s^{(0)}$, -4.8, push cart to the right ),
( $s^{(1)}$, 100, push cart to the right ),
( $s^{(2)}$, -0.42, push cart to the left ),
( $s^{(3)}$, -100, push cart to the left ). |
| Task20 | Acrobot | ( $s^{(3)}$, -1, apply -1 torque ),
( $s^{(4)}$, 12.57, apply 0 torque ),
( $s^{(5)}$, 28.27, apply 1 torque ). |
| Task21 | MountainCar | ( $s^{(0)}$, -0.07, not accelerate ), ( $s^{(1)}$, 0.07, accelerate to the right ). |
| Task22 | Pendulum | ( $s^{(2)}$, 8, maximum left torque ), ( $s^{(1)}$, -1, maximum right torque ). |
| Task23 | Pendulum | ( $s^{(2)}$, 8, maximum left torque ), ( $s^{(2)}$, -8, maximum right torque ). |
| Task24 | Pendulum | ( $s^{(1)}$, -1, maximum right torque ), ( $s^{(2)}$, -8, maximum right torque ). |
| Task25 | Pendulum | ( $s^{(2)}$, 8, maximum left torque ),
( $s^{(1)}$, -1, maximum right torque ),
( $s^{(2)}$, -8, maximum right torque ). |
| Task26 | Lunar Lander | ( $s^{(0)}$, 1.5, do nothing ). |
| Task27 | Lunar Lander | ( $s^{(2)}$, -5, fire left orientation engine ). |
| Task28 | Lunar Lander | ( $s^{(4)}$, 3.14, fire main engine ). |
| Task29 | Lunar Lander | ( $s^{(6)}$, 0, fire right orientation engine ). |
| Task30 | Bipedal Walker | ( $s^{(0)}$, 3.14, maximum forward speed ). |
| Task31 | Bipedal Walker | ( $s^{(1)}$, 5, maximum backward speed ). |
| Task32 | Lunar Lander | ( $s^{(0)}$, 1.5, do nothing ), ( $s^{(4)}$, 3.14, fire main engine ). |
| Task33 | Lunar Lander | ( $s^{(2)}$, -5, fire left orientation engine ), ( $s^{(6)}$, 0, fire right orientation engine ). |
| Task34 | Lunar Lander | ( $s^{(0)}$, 1.5, do nothing ), ( $s^{(6)}$, 0, fire right orientation engine ). |
| Task35 | Lunar Lander | ( $s^{(2)}$, -5, fire left orientation engine ), ( $s^{(4)}$, 3.14, fire main engine ). |
| Task36 | Lunar Lander | ( $s^{(0)}$, 1.5, do nothing ),
( $s^{(2)}$, -5, fire left orientation engine ),
( $s^{(4)}$, 3.14, fire main engine ),
( $s^{(6)}$, 0, fire right orientation engine ). |
| Task37 | Bipedal Walker | ( $s^{(0)}$, 3.14, maximum forward speed ), ( $s^{(1)}$, 5, maximum backward speed ). |
| Task38 | Half Cheetah | ( $s^{(1)}$, 5, apply a torque of 1 to all rotors ). |
| Task39 | Half Cheetah | ( $s^{(2)}$, 5, apply a torque of -1 to all rotors ). |
| Task40 | Hopper | ( $s^{(1)}$, 5, apply a torque of 1 to all rotors ). |
| Task41 | Hopper | ( $s^{(2)}$, -5, apply a torque of -1 to all rotors ). |
| Task42 | Reacher | ( $s^{(0)}$, 5, apply a torque of 1 to all rotors ). |
| Task43 | Reacher | ( $s^{(1)}$, -5, apply a torque of -1 to all rotors ). |
| Task44 | Half Cheetah | ( $s^{(1)}$, 5, apply a torque of 1 to all rotors ), ( $s^{(2)}$, 5, apply a torque of -1 to all rotors ). |
| Task45 | Hopper | ( $s^{(1)}$, 5, apply a torque of 1 to all rotors ), ( $s^{(2)}$, -5, apply a torque of -1 to all rotors ). |
| Task46 | Reacher | ( $s^{(0)}$, 5, apply a torque of 1 to all rotors ), ( $s^{(1)}$, -5, apply a torque of -1 to all rotors ). |
| Task47 | Predator-Prey | ( $s^{(4)}$, 0, move left at max speed ). |
| Task48 | Predator-Prey | ( $s^{(5)}$, 0, remain in place ). |
| Task49 | WorldCom | ( $s^{(4)}$, 0, move left at max speed ). |
| Task50 | WorldCom | ( $s^{(5)}$, 0, remain in place ). |

