# OpenReview forum: "Angel or Demon: Investigating the Plasticity-Enhanced Strategies' Impact on Backdoor Threats in Deep Reinforcement Learning"
_ICLR.cc/2026/Conference — Submitted to ICLR 2026_

### Official Review · Reviewer_Y4ax · 2025-10-28

**Soundness:** 2
**Presentation:** 2
**Contribution:** 2
**Rating:** 4
**Confidence:** 5

**Summary:**

The paper performs a study on the impact of standard Machine Learning interventions on the effectiveness of backdoor attacks in deep reinforcement learning. Specifically, they measure changes in attack success rate and benign episodic return metrics in DRL agents targeted by backdoor attacks before and after interventions like Weight Clipping, Layer Normalization, or Weight Decay are applied. They find that these interventions generally do not impact the performance of backdoor attacks like TrojDRL, UniDoor, and BadRL when the agent is poisoned during its initial training (TM1), but they do have non-trivial impact when agents are poisoned in a post-training regime (TM2).

The motivation of this is to study whether backdoor attacks can be prevented or made more easily detectable with the implementation of standard ML interventions.

**Strengths:**

* I generally really appreciate papers like this that take a scientific approach towards understanding more subtle or ignored aspects of AI security. I think the authors' chosen angle of studying standard ML interventions is reasonable and fairly well motivated.

* Section 5, in particular Figure 5c, is interesting and scientific, though it is hard to interpret as I will discuss later.

* The paper's experimental results are extensive with respect to their chosen attacks, threat models, and interventions

* The proof in Appendix E, despite its assumptions, is appreciated.

**Weaknesses:**

### Misguided Conclusions

1. I think this paper is misguided in trying to find a striking result in the data rather than analyzing and appreciating the results for what they are. Specifically, based upon the experimental results, I disagree with the author's claim that these interventions "mitigate backdoor attacks" [abstract]. Figure 3 along with the numerical results in the appendix indicate to me a very minimal impact of these interventions on the effectiveness of backdoor attacks. Instead, given that both the agent's ASR and BTP scores decreased across nearly all interventions, I think it is more likely that the interventions simply resulted in a universal degradation of performance, which included a decrease in performance in triggered states. Therefore, I don't think these interventions isolate backdoor attacks and decrease their effectiveness. I think practitioners may also avoid implementing these interventions if they truly result in decreased performance.

2. Evaluating the impact of these interventions in the TM2 setting, where the adversary post-trains an existing model to inject the backdoor, seems strange to me. My understanding is that these interventions must be implemented at training time, meaning the adversary is the one using, for instance, weight clipping. Am I correct in this assessment? If this is the case, and given that the interventions decrease BTP performance, I find it unlikely that an adversary would willingly use these interventions.

### Incomplete Evaluation

To be clear, as I mentioned in my Strengths section, the paper's results are very extensive within the scope of the 3 attacks and 9 environments chosen by the authors. Studying over 10,000 combinations of attacks, environments, and threat models is impressive and appreciated. That being said, I think the authors missed an opportunity in (1.) only evaluating attacks using static reward poisoning (TrojDRL, BadRL, and UniDoor all use the same fundamental reward poisoning strategy), and (2.) only exploring continuous action space domains.

1. I think these results would be more compelling if they showed promising mitigations against more state of the art backdoor attack strategies like SleeperNets, which the authors already cite. I do have sympathy for the authors here if they began or completed their evaluations before SleeperNets was published, as their experimental results are surely time consuming, but their results are incomplete without analyzing it in my opinion. Furthermore, since SleeperNets has not been evaluated in continuous action space domains before, to the best of my knowledge, this could be a small contribution on its own.

2. In regards to environments, it is my opinion that the paper would benefit from further evaluations on discrete action space domains, like Atari, in lieu of evaluations on simpler, classic control tasks like cart pole. That being said, it a reasonable restriction given the scope of the results.

### Presentation of Results

1. In general it seems there is a disconnect between the way results are explained in the text versus how they appear in the plots and figures. For instance, in the text around figure 5 a 39.31% change in weight magnitude "performance ranges" is claimed, however in the plot there appears to be little difference between the conventional training versus backdoor training results. To me this misunderstanding stems from the fact that the text is seemingly assessing variance in results, noting that backdoor models result in higher variance, while readers looking at the plots will be primarily concerned with mean values. Perhaps the authors can find more clearly insightful results if they analyze some results on a per-case basis (e.g. noting a particular environment or attack with a large discrepancy), rather than looking at aggregates. This may give more insight into particular cases where these interventions are impactful, rather than seeing the very small changes observed in aggregate.

2. Continuing with section 5.2, figure 6 is very confusing and hard to interpret in my judgement. I think some of the confusion stems from the "rank" metric being used and its conflict with the term "rank" used in linear algebra and throughout the paper. Additionally, in my mind, a lower rank usually indicates higher impact (e.g. someone who ranks #1 in a race is the winner), while here a rank of 8 seems to be the highest score. Perhaps a term like "relative impact" would be more clear.

### Minor Weaknesses

* I don't think the threshold $\epsilon$ is ever given a value.

* Many figures are very small and hard to read. I think sections like the Combination Analysis can be moved to the appendix to create more space for larger figures and more analysis.

* The legend in figure 4 clips over some of the results

### Conclusion

Overall I think the results of this paper are too inconclusive. I do appreciate the authors' efforts, and I do believe that such evaluations are important, but I'm unsure if the claimed results are significant enough in their mitigation of backdoor attacks against DRL to warrant publication in a major ML venue as a finished product. I believe there are likely other interesting conclusions which may be drawn from the data, but the current paper does not bring these to light in its current form.

**Questions:**

* What value of $\epsilon$ is used in this paper's evaluation?

* When evaluating TM2 are the interventions performed during the adversary's post-training phase?

* When evaluating BTP with interventions, are the authors comparing against a "Conventional Training" agent trained with or without interventions?

---

> ### Author Response · Authors · 2025-11-25
> **Response to Reviewer Y4ax**
>
> Hello Reviewer Y4ax,
>
> We sincerely appreciate your recognition of the motivation of this study and the scale of our experiments.
> Additionally, we greatly appreciate your detailed comments, which are both thorough and meaningful.
> Based on your feedback, we have improved the manuscript by adding a novel backdoor attack SleeperNets, conducting a supplementary investigation in MPE, refining the presentation, and providing more detailed explanations of the experimental settings and results.
> In the following, we reply to the questions one by one for the convenience of checking.
> We apologize for the length of this rebuttal, which was necessary to address the many questions raised.
> We hope that our responses address your concerns and look forward to further discussion with you.
> Meanwhile, we would be grateful if you might consider increasing your rating.
>
> ---
>
> **Misguided Conclusions 1 & Q3**: Figure 3 and the numerical results in the appendix suggest that these interventions have only a minimal impact on backdoor effectiveness. Since both ASR and BTP decrease under nearly all interventions, it appears more likely that the interventions cause a general degradation of performance. When evaluating BTP with interventions, are the authors comparing against a "Conventional Training" agent trained with or without interventions?
>
> **A1**: Thanks for your suggestion.
> We apologize for not emphasizing this point, which may have caused confusion.
> * In fact, the original manuscript already investigated that **interventions in the Conventional Training setting have negligible effect on the BTP of DRL agents**.
> Therefore, interventions do indeed lead to a decrease in BTP during DRL backdoor attacks, particularly in *TM-Post*.
> * In the revised manuscript, we further emphasize the statement in Section 5.1:
> "**Appendix F** shows that these interventions have negligible impact on BTP under conventional training. Therefore, the fluctuations in BTP observed in Figure 3(b) are caused by the effects of interventions on DRL backdoor attacks."
> Some of the results are as follows:
>
> | Intervention         | BTP (mean ± standard deviation) | Difference |
> |---------------------|--------------------------------|-----------|
> | Conventional Training              | 0.989 ± 0.032                  | +0.000    |
> | *Shrink & Perturb*  | 0.991 ± 0.025                  | +0.002    |
> | *Layer Norm*        | 0.985 ± 0.074                  | -0.004    |
> | *Weight Clipping*   | 0.982 ± 0.070                  | -0.007    |
> | *Spectral Norm*     | 0.983 ± 0.027                  | -0.006    |
> | *Weight Decay*      | 0.987 ± 0.036                  | -0.002    |
> | *ReDo*              | 0.979 ± 0.055                  | -0.012    |
> | *SAM*               | 0.977 ± 0.068                  | -0.010    |
>
> ---
>
> **Misguided Conclusions 2 & Q2**: When evaluating *TM2* are the interventions performed during the adversary's post-training phase?
>
> **A2**: Thank you for pointing out this crucial issue.
> We have added a statement addressing this issue in **Appendix B** of the revised manuscript.
> Specifically, we respond to your concerns in the following two steps：
>
> *Concern 1. In *TM-Post*, is the adversary required to strictly adhere to the interventions embedded by the provider?*
>
> * In *TM-Post*, the adversary is not necessarily constrained to follow the interventions embedded by the provider.
> For example, the adversary is allowed to remove interventions such as *Weight Clipping* or *ReDo*, which has a negligible impact on post-training.
> **In contrast**, the removal of interventions such as *Spectral Normalization* or *Layer Normalization* is generally infeasible, as it is prone to induce substantial performance degradation or even catastrophic failure of the DRL agent during post-training.
>
> *Concern 2. If the adversary is not constrained to adhere to the provider's embedded interventions, does investigating this scenario remain meaningful?*
>
> * Considering *TM-Post* is essential, and it constitutes one of the primary motivations of this study.
> In existing studies, the adversary remains unaware of whether the interventions influence DRL backdoor attacks.
> **Since one of the adversary's goals is to preserve the victim agent's performance on benign tasks (i.e., BTP), there is an incentive to retain the provider's embedded interventions, or even to introduce additional interventions to compensate for BTP degradation caused by the backdoor attack.**
> Therefore, the adversary might overlook the fact that certain interventions could either exacerbate or mitigate the backdoor threat.
> **This study provides insights for both the adversary and the provider/defender**: the adversary leverages these insights to exacerbate backdoor threats, while the provider/defender uses them to mitigate such threats.

---

> ### Author Response · Authors · 2025-11-25
> **Response to Reviewer Y4ax (2)**
>
> **Incomplete Evaluation**: I think the authors missed an opportunity in (1.) only evaluating attacks using static reward poisoning (TrojDRL, BadRL, and UniDoor all use the same fundamental reward poisoning strategy), and (2.) only exploring continuous action space domains.
>
> **A3**: Sorry for the confusion.
>
> * Clarification 1: TrojDRL and BadRL employ static reward poisoning, while UNIDOOR uses dynamic reward poisoning, in which the backdoor reward is continuously adjusted based on the agent's performance on both benign and backdoor tasks.
> Nevertheless, we also believe that incorporating **SleeperNets** is beneficial, and thus we have included it in the revised manuscript.
> We incorporated action tampering into SleeperNets to improve its effectiveness in continuous action spaces.
> We believe that better hyperparameter settings could further enhance the attack performance of SleeperNets.
> Nevertheless, considering the large scale of our experiments, we kept the reward constant fixed at 5 and the weighting factor at 0.5.
> This process is very time-consuming, which is the main reason for our delayed response to the rebuttal.
> All relevant results have been updated in the revised manuscript.
>
> * Clarification 2: Our investigation actually includes continuous action tasks, and we simultaneously considered four other dimensions to cover a wide range of task types.
> The following table provides a summary.
> For a more detailed analysis of the DRL tasks, please refer to **Appendix E** in the revised manuscript.
>
> | Categories                 | DRL Tasks      | Action Space | Action Dim. | Reward Type | Reward Normalization | Cold-Start |
> |----------------------------|----------------|--------------|-------------|-------------|------------|------------|
> | **Classic Control Tasks**  | CartPole       | Discrete     | 1D          | Dense       | ×          | ×          |
> |                            | Acrobot        | Discrete     | 1D          | Sparse      | ×          | ×          |
> |                            | MountainCar    | Discrete     | 1D          | Sparse      | ×          | ✔          |
> |                            | Pendulum       | Continuous   | 1D          | Dense       | ×          | ×          |
> | **Physics Control Tasks**  | Lunar Lander   | Discrete     | 1D          | Dense       | ×          | ×          |
> |                            | Bipedal Walker | Continuous   | N-D         | Dense       | ×          | ✔          |
> | **Robotic Tasks**          | Hopper         | Continuous   | N-D         | Dense       | ✔          | ×          |
> |                            | Reacher        | Continuous   | N-D         | Dense       | ✔          | ×          |
> |                            | Half Cheetah   | Continuous   | N-D         | Dense       | ✔          | ✔          |
>
> Nevertheless, we agree that extending the range of tasks is reasonable.
> Therefore, we conducted a supplementary investigation, which is designed to include two multi-agent competitive tasks in the **MPE** environment (Predator-Prey and WorldCom), two classical DRL algorithms (**DDPG** and **MADDPG**), four backdoor tasks, UNIDOOR as the attack method, and three intervention settings (*None*, *Layer Normalization*, and *SAM*).
> The following presents the overall results in post-training scenarios (i.e., *TM-Post*).
> More details on the design, results, and analysis can be found in **Appendix J** of the revised manuscript.
>
> Intervention        | Predator-Prey (ASR ↑) | Predator-Prey (BTP ↑) | WorldCom (ASR ↑) | WorldCom (BTP ↑) |
> |------------------|-------------------|-------------------|----------------|----------------|
> |*None*             | 0.202             | 1.000             | 0.255          | 0.962          |
> |*Layer Normalization* | 0.209           | 0.887             | 0.242          | 0.923          |
> |*SAM*              | 0.319             | 0.993             | 0.409          | 0.960          |

---

> ### Author Response · Authors · 2025-11-25
> **Response to Reviewer Y4ax (3)**
>
> **Presentation of Results 1**: It seems there is a disconnect between the way the results are explained in the text and how they appear in Figure 5.
>
> **A4**: Thanks for your suggestion.
> Your understanding is correct.
> That is, we aim to highlight the fluctuations in the pathological characteristics of the backdoor attack, in order to emphasize the anomaly in the loss landscape sharpness.
> We have revised the manuscript (Section 5.2) to avoid potential misunderstandings:
> * The performance ranges (i.e., the absolute differences between the maximum and minimum results) of weight magnitude and effective rank increase by 98.63% and 19.16%, respectively, with the most pronounced effect observed in loss landscape sharpness, whose range increases by 635.22%.
>
> ---
>
> **Presentation of Results 2**: Figure 6 is very confusing and hard to interpret in my judgement.
>
> **A5**: Thanks for your suggestion.  "Relative impact" is a good alternative, but after discussion, we decided to use "ranking" in the revised manuscript to avoid confusion with the concept of "rank" in linear algebra, as it is more precise.
> Additionally, we have added a part titled **Ranking Criteria** in **Appendix G** to explain how we rank the interventions based on different pathological characteristics:
>
> * **Ranking Criteria**. Aligned with prior plasticity studies, for the pathological characteristics weight magnitude and loss landscape sharpness, smaller values correspond to higher intervention rankings, whereas for effective rank, larger values correspond to higher rankings. The highest rank is 1 and the lowest is 8, meaning that the average ranking ranges from 1 to 8. For example, in Figure 14(a), *Weight Clipping* is ranked \#1 for Task0, indicating that in this backdoor attack scenario, compared with other intervention settings, it reduces the weight magnitude to the lowest value.
>
> ---
>
> **Minor Weaknesses 1 & Q1**: What value of $\epsilon$ is used in this paper's evaluation?
>
> **A6**: Apologies for the omission of this detail.
> We have added this information in **Appendix E** of the revised manuscript.
> Specifically, following the setup in [1], we set $\epsilon$ to 0.1 for Bipedal Walker, Predator-Prey, and WorldCom, and to 0.05 for the other tasks.
>
> [1] UNIDOOR: A Universal Framework for Action-Level Backdoor Attacks in Deep Reinforcement Learning.
>
> ---
>
> **Minor Weaknesses 2**: Many figures are very small and hard to read. I think sections like the Combination Analysis can be moved to the appendix to create more space for larger figures and more analysis.
>
> **A7**: Thanks for your suggestion.
> In the revised manuscript, we have reconstructed all figures.
> Specifically, we have increased the space occupied by each figure in the manuscript and enlarged the font size of all text within the figures.
> Additionally, we have split Figure 10 into Figures 10 and 18, with the latter being moved to the Appendix.
>
> ---
>
> **Minor Weaknesses 3**: The legend in figure 4 clips over some of the results.
>
> **A8**: Thanks for your suggestion.
> In the revised manuscript, we have adjusted the presentation of all figure legends to more clearly convey the results.

---

> > ### Comment · Reviewer_Y4ax · 2025-11-25
> > **Response to Authors**
> >
> > Thank you for your detailed rebuttal and improvemenmts made to the paper. I believe the quality of the paper has improved significantly. Specifically, it is now much easier to understand the specific scientific questions this paper is exploring along with the findings that have been made, so I appreciate that.
> >
> > Overall I would support this paper for publication as a poster unless other reviewers convince me otherwise.

---

### Official Review · Reviewer_q1rd · 2025-10-31

**Soundness:** 3
**Presentation:** 2
**Contribution:** 2
**Rating:** 4
**Confidence:** 2

**Summary:**

This paper studies backdoor attacks against Deep Reinforcement Learning (DRL). In real-world deployments, DRL agents often adopt intervention strategies (e.g., normalization, weight regularization, sharpness control) to improve stability and mitigate plasticity loss. However, no prior work has examined how these interventions themselves influence the effectiveness of backdoor attacks. The goal of this paper is therefore to investigate how and why intervention strategies affect the behavior of DRL backdoor attacks.

To this end, the authors conduct over ten thousand experiments under two threat models (training-time and post-training injection), eight types of interventions, and three representative backdoor attack methods. This paper first evaluates how different interventions impact the attack success rate (ASR) and benign task performance (BTP), showing that some interventions mitigate attacks while others not. Then, they analyze the underlying causes through four pathological indicators (weight magnitude, effective rank, loss-landscape sharpness, dormant neurons), and condlude that interventions influence the model’s adaptability and stability. Finally, they explores how combinations of interventions affect backdoor.

**Strengths:**

1. The paper addresses an important and previously unexplored question: how intervention strategies affect backdoor attacks. This perspective is novel because most prior works focus on improving DRL performance or robustness, while overlooking the potential security side effects of such interventions.


2. The paper conducts a large number of experiments covering diverse DRL tasks, attack settings, and intervention strategies. The experimental scale and breadth are sufficient to provide a solid empirical foundation and can serve as a valuable reference for future work in this area. Although certain concerns about the experimental depth will be mentioned later, the comprehensiveness of the evaluation remains one of the paper’s clear strengths.

**Weaknesses:**

1. Although the paper provides extensive experimental results, many of the conclusions are not sufficiently verified through follow-up analysis. For example, in Section 5.2 the authors claim that SAM strengthens backdoor attacks because it makes the loss landscape smoother. However, they do not perform additional experiments (for instance, by making the loss landscape sharper to see if the attack weakens), which would be necessary to clearly validate this explanation. As a result, although most of the conclusions appear reasonable, they are not yet convincing enough to be fully trusted due to the lack of sufficient validation.

2. The paper reports many findings across different interventions and metrics, but these results are presented in a fragmented way without forming a coherent main message. Although the paper highlights two key insights in the conclusion: (1) Robust Backdoor Injection (2) Sharpness-Based Detection. While both are interesting and practically useful observations, they seems to application-level insights rather than general principles. As a result, the reader may find it difficult to identify the key takeaway.

**Questions:**

1. What is the key conclusion or principle that readers should take away? (explaination see Weakness #2)

2. Some of the conclusions in the paper seem well supported by evidence, while others appear more preliminary.
Could the authors clarify which findings they consider most reliable and which ones are more exploratory or require further validation?

---

> ### Author Response · Authors · 2025-11-25
> **Response to Reviewer q1rd**
>
> Hello Reviewer q1rd,
>
> We greatly appreciate your recognition of the motivation of this study and concur that it offers a solid empirical foundation.
> In addition, we sincerely thanks for your insightful suggestions.
> In response to these suggestions, we have refined and highlighted the findings in the revised manuscript, strengthened their analysis, and clarified their logical relationships.
> In the following, we reply to the questions one by one for the convenience of checking.
> We hope that our responses address your concerns and help reinforce your confidence in this study.
> We also welcome any additional comments and would be grateful if you might consider increasing your rating.
>
>
> ---
>
> **W1**: The claim that *SAM* strengthens backdoor attacks requires further justification.
>
> **A1**: Thanks for your insightful suggestion.
> We agree that further exploration of the impacts of *SAM* on backdoors is necessary, and we have made the following two improvements in the revised manuscript:
> * A more detailed explanation of the **intrinsic mechanisms** has been added in Section 5.2: "Its intrinsic mechanism is that *SAM* captures the backdoor direction via the sharp losses and amplifies the corresponding gradients, enabling the backdoor pathway to rapidly converge into a flat-minimum region that is robust to parameter perturbations.
> This reduces continuous competition between backdoor and benign pathways in the inherently non-stationary training process of DRL.
> The effect is especially pronounced in *TM-Post*, where reduced flexibility in the agent's parameter space hinders the formation of the backdoor pathway."
> * We have emphasized the **theoretical proof** presented in **Appendix I**: "Appendix I presents a theoretical proof, leveraging influence functions [2] to demonstrate how *SAM* amplifies backdoor threats."
>
> [1] "Sharpness-Aware Minimization for Efficiently Improving Generalization", ICLR.
>
> [2] "Understanding black-box predictions via influence function", ICML.

---

> ### Author Response · Authors · 2025-11-25
> **Response to Reviewer q1rd (2)**
>
> **W2 & Q1**：What is the key conclusion or principle that readers should take away?
>
> **A2**: Thanks for your insightful suggestion.
> Inspired by your suggestion, we have refined and highlighted the **five main findings in the Section 5** of the revised manuscript.
> Below, we outline the content of these findings and elaborate on the relationships between them:
>
> * **Finding 1**. In TM-Post, interventions exert a more substantial impact on DRL backdoor attacks than in *TM-Scratch*. Notably, SAM exacerbates the backdoor threat, whereas the other interventions exhibit varying degrees of mitigation.
>
> * **Finding 2**. Interventions characterized by noise, clipping, and reset (e.g., *Shrink \& Perturb*, *Weight Clipping*, and *ReDo*) disrupt activation pathways, leading to competitive reconstruction between backdoor and benign pathways.
>
> * **Finding 3**. Interventions that compress the representation space shift the gradient directions of backdoor and benign states from orthogonal to aligned (e.g., *Spectral Normalization*, *Weight Decay*, and *Layer Normalization*), transforming backdoor pathways from sparse to dense.
>
> * **Finding 4**. *SAM* amplifies the backdoor gradients and enables the backdoor pathway to rapidly converge while remaining robust to parameter perturbations, thereby alleviating competition between backdoor and benign tasks---a phenomenon especially pronounced in *TM-Post*.
>
> * **Finding 5**. In *TM-Post*, interventions whose intrinsic mechanisms involve disrupting activation pathways and compressing the representation space may act as catalysts for DRL backdoor attacks when combined with *SAM*.
>
> **Coherence**.
> **Finding 1 reveals that interventions have a pronounced impact in the post-training stage.**
> **Findings 2–4 reveal three intrinsic mechanisms through which interventions affect DRL backdoor attacks, and Finding 5 demonstrates that these three intrinsic mechanisms can be systematically combined.**
> **Based on these findings, we propose the *SCC* framework to support robust backdoor attacks in the post-training scenario, with its three key components designed to correspond to the intrinsic mechanisms outlined above.**
>
> ---
>
> **Q2**：Could the authors clarify which findings they consider most reliable and which ones are more exploratory or require further validation?
>
> **A3**: Thanks for your suggestion.
> This study summarizes the five findings described above, along with two insights: *Robust Backdoor Injection* and *Sharpness-Based Detection*.
> We believe Findings 1–4 and *Sharpness-Based Detection* are fully reliable, while Finding 5 and *Robust Backdoor Injection* remain exploratory and warrant further validation due to their complexity.
> This arises from:
>
> * We find that interventions with different intrinsic mechanisms can potentially be combined to amplify backdoor threats, but it remains unclear exactly how they should be combined. We believe that the proposed *SCC* framework provides high-level guidance, though further fine-grained investigation is needed.
>
> * We have analyzed the intrinsic mechanisms through which each intervention impacts DRL backdoor attacks, yet how these mechanisms interact when interventions are combined remains unclear. We proposed one interesting approach (i.e., the **Pathological Diagnosis** in Section 5.3), but it still warrants further exploration.
>
> Therefore, we describe *SCC* as a **conceptual framework** in the manuscript.
> We hope that *SCC* will inspire future research aimed at uncovering more robust DRL backdoor attacks in the post-training scenario, which is crucial for assessing the security of real-world DRL deployments.

---

### Official Review · Reviewer_Gy5W · 2025-10-31

**Soundness:** 3
**Presentation:** 1
**Contribution:** 3
**Rating:** 6
**Confidence:** 3

**Summary:**

This paper tries to understand the intersection of two fields of reinforcement learning study: techniques used to maintain plasticity to allow for long term reinforcement learning, and attacks designed to insert backdoor behavior in response to certain observational triggers. They seek to understand how the application of different plasticity techniques impacts the feasibility of successfully executing backdoor attacks, and find that most common techniques make backdoor training harder, while only one (SAM) improves attack success rate. The authors do this across two attack settings: one where an attack is inserted during training, and another where it is inserted during post-training. They then go on to try to understand commonalities between techniques that make backdoor creation harder, and identify that multiple techniques have the effect of increasing the rank of weight matrices. They believe that this effective "promotion" of smaller-magnitude directions helps drown out the salience of the trigger used in the backdoor.

**Strengths:**

- I appreciated the systematic testing of multiple interventions over multiple backdoor threat models
- The discussion section gave valuable insights into why we might expect the interventions studied to have the effects they do on backdoor injection

**Weaknesses:**

- Plots and figures were tiny relative to the scale of text, making it hard to see and understand them
- Writing quality was generally poor and made the paper hard to understand
- The use of shortened acronyms (e.g. TM1 and TM2 for pretraining and posttraining injection) made the paper hard to parse - I had to keep reminding myself which is which. The paper would have been much easier to read if each acronym were replaced with a (shortened) reference to what it was actually referring to

**Questions:**

- How is it possible for SAM to induce a greater than 100% Success Attack Rate? This was quite confusing, and made me wonder if there was a typo

---

> ### Author Response · Authors · 2025-11-25
> **Response to Reviewer Gy5W**
>
> Hello Reviewer Gy5W,
>
> We are grateful for your recognition of the two insights derived from our systematic investigation.
> In addition, we sincerely thanks for your insightful suggestions.
> In response, we have improved the presentation by highlighting our findings, clarifying the logical flow, and revising the figures.
> In the following, we reply to the questions one by one for the convenience of checking.
> We hope our responsees address your concerns and increase your confidence in our paper and results.
> We welcome any further comments you may have, and would greatly appreciate it if you could consider increasing your rating.
>
> ---
>
> **W1**: Plots and figures were tiny relative to the scale of text, making it hard to see and understand them.
>
> **A1**: We apologize for any inconvenience caused during your reading.
> In the revised manuscript, we have reconstructed all plots and figures:
>
> * We increased the space occupied by each figure in the manuscript and enlarged the font size of all text within the figures.
> * We adjusted the legend placement to avoid obstructing the results.
> * We split Figure 10 into Figures 10 and 18, with the latter moved to the Appendix.
>
> ---
>
> **W2**：The use of shortened acronyms (e.g. *TM1* and *TM2* for pretraining and posttraining injection) made the paper hard to parse.
>
> **A2**: Thanks for your suggestion.
> In the revised manuscript, we have made the following improvements:
> * *TM1* and *TM2* are replaced with *TM-Scratch* and *TM-Post*, corresponding to the learn-from-scratch and post-training threat models, respectively.
> * We reduced some symbols to improve readability, such as R1–R3 in Section 5.2 and C1–C3, S1, and S2 in Section 5.3.
> * We ensured that all shortened acronyms are defined at their first occurrence.
>
> ---
>
> **W3**: Writing quality was generally poor and made the paper hard to understand.
>
> **A3**: Thanks for your insightful suggestion.
> We have improved the writing to emphasize key points, enhancing clarity and conciseness.
> Most importantly, we have summarized and highlighted **five findings** in the Section 5 of the revised manuscript:
>
> * **Finding 1**. In *TM-Post*, interventions exert a more substantial impact on DRL backdoor attacks than in *TM-Scratch*. Notably, SAM exacerbates the backdoor threat, whereas the other interventions exhibit varying degrees of mitigation.
>
> * **Finding 2**. Interventions characterized by noise, clipping, and reset (e.g., *Shrink \& Perturb*, *Weight Clipping*, and *ReDo*) disrupt activation pathways, leading to competitive reconstruction between backdoor and benign pathways.
>
> * **Finding 3**. Interventions that compress the representation space shift the gradient directions of backdoor and benign states from orthogonal to aligned (e.g., *Spectral Normalization*, *Weight Decay*, and *Layer Normalization*), transforming backdoor pathways from sparse to dense.
>
> * **Finding 4**. *SAM* amplifies the backdoor gradients and enables the backdoor pathway to rapidly converge while remaining robust to parameter perturbations, thereby alleviating competition between backdoor and benign tasks---a phenomenon especially pronounced in *TM-Post*.
>
> * **Finding 5**. In *TM-Post*, interventions whose intrinsic mechanisms involve disrupting activation pathways and compressing the representation space may act as catalysts for DRL backdoor attacks when combined with *SAM*.
>
> **Coherence**.
> **Finding 1 reveals that interventions have a pronounced impact in the post-training stage.**
> **Findings 2–4 reveal three intrinsic mechanisms through which interventions affect DRL backdoor attacks, and Finding 5 demonstrates that these three intrinsic mechanisms can be systematically combined.**
> **Based on these findings, we propose the *SCC* framework to support robust backdoor attacks in the post-training scenario, with its three key components designed to correspond to the intrinsic mechanisms outlined above.**
>
> ---
>
> **Q1**: How is it possible for SAM to induce a greater than 100% Attack Success Rate?
>
> **A4**: Sorry for the confusion.
> In the manuscript, we report relative performance improvements, as ASR and BTP are measured on a scale from 0.000-1.000 rather than 0%–100%.
> For example, in the post-training scenario, *SAM* increases the average ASR (i.e., attack success rate) of TrojDRL in Robotics Tasks from 0.344 to 0.778 (as shown in Table 7 of the revised manuscript), resulting in a relative performance improvement of (0.778−0.344) / 0.344 = 126.16%.
> We have provided a detailed example in Section 1 of the revised manuscript to prevent potential confusion for readers.

---

### Official Review · Reviewer_bCoi · 2025-11-01

**Soundness:** 1
**Presentation:** 1
**Contribution:** 1
**Rating:** 2
**Confidence:** 4

**Summary:**

This paper focuses on backdoor attacks and deep reinforcement learning plasticity, and  shows that interventions, such as Weight Clipping and Layer Normalization, mitigate backdoor attacks by expanding the agent’s representation space.

**Strengths:**

Please see below.

**Weaknesses:**

It is already well-known that weight clipping and layer normalization mitigates backdoor attacks [1,2,3]. These concepts have been well studied. None of these prior work on demonstrating the exact same results were discussed, cited or mentioned even in the background. These issues are foundational to deep neural networks and well-studied. There is nothing specific to reinforcement learning that makes the results even remotely surprising.

I also find it quite concerning that the paper did not read and review prior work and try to place their claims and contributions according to that. There is a long list of citations, i.e. 5 pages, yet this long list somehow excludes the papers that are showing quite similar issues.

Only one single deep reinforcement learning algorithm is tested. This is not enough evidence to convey general claims about deep reinforcement learning.

Most of the experimental section is based on quite simple tasks such as CartPole, Acrobot,  MountainCar, Lunar Lander with discrete action set and low dimensional states. These are not the environments where a publication can make general claims about deep reinforcement learning. The paper studies concepts that are basic to deep reinforcement learning. Yet, the experimental setting does not reflect a setting that can verify the claims made in the paper regarding deep RL.

Furthermore, looking at the results, TM1 is always within the standard deviation across interventions. Again this significantly limits the claims of the paper.

I also disagree with the phrase “DRL agents typically incorporate intervention strategies,
which mitigates the plasticity loss problem”. This is just a new research area, it has not much to do with what happens in practice.



Essentially, only 3 backdoor attacks have been used [4,5,6]. But the results are reported to inflate and make the impression that a comprehensive study has been conducted. I find this kind of presentation quite misleading.

There are many more backdoor attacks in deep reinforcement learning. I am not necessarily going to cite all of them here. But I just find it misleading to constantly phrase working on 10s of thousands cases while essentially working on simply 3 backdoor attacks.

Given all these limitations, this paper might be more appropriate for a venue that will appreciate slightly more known and expected results.


[1] Can You Really Backdoor Federated Learning?, NeurIPS 2019.

[2] A Gradient Control Method for Backdoor Attacks on Parameter-Efficient Tuning, ACL 2023.

[3] Defense against Backdoor Attack on Pre-trained Language Models via Head Pruning and Attention Normalization, ICML 2024.

[4] BadRL: Sparse Targeted Backdoor Attack against Reinforcement Learning, AAAI 2024.

[5] UNIDOOR: A Universal Framework for Action-Level Backdoor Attacks in Deep Reinforcement Learning, Arxiv 2025.

[6] TrojDRL: Evaluation of Backdoor Attacks on Deep Reinforcement Learning, ACM/IEEE Design Automation Conference (DAC) 2020.

**Questions:**

Please see above.

---

> ### Author Response · Authors · 2025-11-25
> **Response to Reviewer bCoi**
>
> Hello Reviewer bCoi,
>
> Thanks for your feedback and constructive suggestions.
> While we acknowledge that this study can be further improved, we believe it offers a novel perspective and provides meaningful insights into DRL backdoor research.
> In response to your comments, we have provided detailed answers, clarified the sources of confusion, and expanded our investigation.
> We apologize for the length of this rebuttal, which was necessary to address the questions raised.
> We sincerely hope that you will carefully consider our responses and re-evaluate the manuscript.
> If you have any further concerns, please let us know, and we will address them promptly.
>
> ---
>
> **W1**: Weight clipping and layer normalization are well-known, extensively studied defenses against backdoor attacks in deep neural networks, and their effects in reinforcement learning are neither novel nor surprising.
>
> **A1**: Sorry for the confusion.
> In fact, our main contribution is to investigate the impacts of seven existing interventions on DRL backdoor attacks and distill three intrinsic mechanisms as follows:
>
> * **M1**: Disrupting activation pathways induces competition between backdoor and benign tasks (e.g., ***Shrink \& Perturb***, ***Weight Clipping***, ***ReDo***).
>
> * **M2**: Compressing the agent's representation space shifts backdoor and benign gradients from orthogonality toward alignment, creating denser backdoor pathways and exacerbating non-stationarity (e.g., ***Spectral Normalization***, ***Weight Decay***, ***Layer Normalization***).
>
> * **M3**: Capturing the backdoor direction via sharp losses and amplifying the corresponding gradients enables the backdoor pathway to rapidly converge into a flat-minimum region, which is robust to parameter perturbations (e.g., ***SAM***).
>
> As shown in Figure 11 of the manuscript, we propose the *SCC* framework to support robust backdoor attacks in the post-training scenario, with its three key components designed to correspond to the intrinsic mechanisms outlined above.
>
> Therefore, rather than analyzing the impact of each intervention on backdoors in isolation, **we systematically examine the intrinsic mechanisms through which they exert their effects, which can be generalized to analyze other interventions.
> We believe it constitutes a clear and substantial contribution to the field.**
>
> In the revised manuscript, we have highlighted the three intrinsic mechanisms and their implications for future research on DRL backdoors.
>
> ---
>
> **W2**: The paper overlooks closely related studies.
>
> **A2**: Thanks for the recommended references. However, we believe their connection to our work is rather limited. For example, the core idea of [1] is that "prune the attention heads that are potentially affected by poisoned texts with only clean texts on hand and then further normalize the weights of the remaining attention heads to mitigate the backdoor impacts." In contrast, our work studies *Layer Normalization* (one of the 7 interventions studied in our work), which performs mean-variance normalization on each sample's hidden states within every layer. This mechanism is fundamentally different from the approach in [1].
>
> Meanwhile, we fully respect existing studies and will briefly discuss how our work relates to them along the dimensions **M1**–**M3** mentioned above:
>
> * **M1** shares similarities with previous studies on DL backdoors but is not identical (we have added a reference to [2] in the revised manuscript).
> We interpret the impact of disrupting activation pathways from the perspective of pathway competition arising during non-stationary DRL training, which may affect the agent's performance on both backdoor and benign tasks.
> In the context of DL backdoor attacks, however, the impact on the model's performance on benign tasks is relatively minor.
>
> * The concept of backdoor orthogonality in **M2** follows [3], which was already cited in the original manuscript.
> On this basis, we reveal that compressing the DRL agent's representation space shifts backdoor and benign gradients from orthogonal to aligned.
>
> * **M3** reveals that DRL backdoors induce abnormal loss landscape sharpness and *SAM* further exacerbates the backdoor threats in post-training scenarios.
> We provided a **theoretical proof in Appendix E** of the original manuscript for why *SAM* exacerbates backdoor threats.
> Meanwhile, we have added a reference to [4] in the revised manuscript.
> [4] discusses the relationship between DL backdoors and parameter space landscapes, which is mechanistically distinct from loss landscape sharpness, yet exhibits certain formal similarities.
>
> [1] "Defense against Backdoor Attack on Pre-trained Language Models via Head Pruning and Attention Normalization", ICML.
>
> [2] "Backdoor Learning: A survey", IEEE TNNLS.
>
> [3] "Exploring the Orthogonality and Linearity of Backdoor Attacks", IEEE S\&P.
>
> [4] "CLIBE: Detecting Dynamic Backdoors in Transformer-Based NLP Models", NDSS.

---

> ### Author Response · Authors · 2025-11-25
> **Response to Reviewer bCoi (2)**
>
> **W3**: TM1 is always within the standard deviation across interventions. Again this significantly limits the claims of the paper.
>
> **A3**: Sorry for the confusion.
>
> * The original manuscript already states that interventions have a significantly greater impact on *TM2* (post-training) than on *TM1* (learn-from-scratch).
> This phenomenon is consistent with observations in plasticity research, where an agent’s plasticity gradually declines during continual learning, thereby amplifying the impact of interventions.
> In the revised manuscript, we have highlighted this observation as **Finding 1** in Section 5.1 of the revised manuscript.
>
> * The statement that "*TM1* is always within the standard deviation across interventions" does not always hold.
> For example, as shown in **Table 5** of the original manuscript, in the Classic Control Tasks, applying *Layer Normalization* reduces the BTP (i.e., benign task performance) of three backdoor attacks from **0.983** to **0.885**.
> Considering that this is an average over 234 cases rather than a single case, this indicates a notable performance decline.
>
> ---
>
> **W4**: I also disagree with the phrase "DRL agents typically incorporate intervention strategies, which mitigates the plasticity loss problem". This is just a new research area, it has not much to do with what happens in practice.
>
> **A4**: Thank you for your suggestion.
> In the revised manuscript, we have corrected the statement to: "plasticity loss is essential to reinforcement learning in ways that go beyond its importance in supervised learning." This assertion, cited from [4], underscores the urgency of interventions in RL. Similar statements are widely reported in the plasticity literature.
>
> [4] "Loss of plasticity in deep continual learning", Nature.
>
> ---
>
> **W5**: Only 3 backdoor attacks have been used. But the results are reported to inflate and make the impression that a comprehensive study has been conducted.
>
> **A5**: Thanks for your suggestion.
> * Our statement that the study is **comprehensive** refers to its full scope, which goes beyond the three DRL backdoor attacks to also include two threat models, eight intervention settings with five combination strategies, and a total of 47 backdoor tasks.
>
> * Meanwhile, we agree that extending the range of attacks is reasonable.
> Therefore, we have incorporated a novel backdoor attack, **SleeperNets** [5], into the revised manuscript, increasing the number of tested cases from 10,998 to 14,664.
> This process is time-consuming, which is the main reason for our delayed response to the rebuttal.
> The results reported in the revised manuscript have now incorporated SleeperNets.
>
> [5] "SleeperNets: Universal Backdoor Poisoning Attacks against Reinforcement Learning Agents", NeurIPS.

---

> ### Author Response · Authors · 2025-11-25
> **Response to Reviewer bCoi (3)**
>
> **W6**: Simple tasks and limited DRL algorithm.
>
> **A6**: We acknowledge that the current study does not cover all possible scenarios, but we firmly believe that it nevertheless spans a sufficiently broad range and that its findings meaningful.
>
> * Tasks:
> In addition to the four discrete-action tasks you mentioned, we also investigated **5 continuous-control tasks**---Hopper, Reacher, HalfCheetah, Pendulum, and BipedalWalker---which are widely regarded as important benchmarks.
> **Table 3** in the manuscript summarizes the characteristics of these tasks, including their discrete or continuous action spaces, one-dimensional or multi-dimensional action outputs, sparse or dense reward types, and whether they exhibit a cold-start issue.
> To the best of our knowledge, some of these tasks remain challenging for current DRL backdoor attacks (e.g., achieving precise target-action control in continuous action spaces).
> In addition, these tasks exhibit consistent environment dependencies, which mitigates environment-conflict issues in DRL research and enhances reproducibility.
>
> | Categories                 | DRL Tasks      | Action Space | Action Dim. | Reward Type | Reward Normalization | Cold-Start |
> |----------------------------|----------------|--------------|-------------|-------------|------------|------------|
> | Classic Control Tasks  | CartPole       | Discrete     | 1D          | Dense       | ×          | ×          |
> |                            | Acrobot        | Discrete     | 1D          | Sparse      | ×          | ×          |
> |                            | MountainCar    | Discrete     | 1D          | Sparse      | ×          | ✔          |
> |                            | Pendulum       | Continuous   | 1D          | Dense       | ×          | ×          |
> | Physics Control Tasks  | Lunar Lander   | Discrete     | 1D          | Dense       | ×          | ×          |
> |                            | Bipedal Walker | Continuous   | N-D         | Dense       | ×          | ✔          |
> | Robotic Tasks          | Hopper         | Continuous   | N-D         | Dense       | ✔          | ×          |
> |                            | Reacher        | Continuous   | N-D         | Dense       | ✔          | ×          |
> |                            | Half Cheetah   | Continuous   | N-D         | Dense       | ✔          | ✔          |
>
> * Algorithm:
> We chose **PPO** for two main reasons:
> (1) PPO is one of the most widely used DRL algorithms.
> Its framework is compatible with nearly all types of tasks, and it exhibits stable performance with relatively low sensitivity to hyperparameters.
> (2) PPO is the algorithm with the greatest overlap between research on DRL plasticity and research on DRL backdoor attacks.
> Considering these two factors and the large scale of our experiments---which span different tasks, plasticity interventions, and attack methods, each introducing additional hyperparameters and potential constraints on algorithmic frameworks---using PPO constitutes the most reasonable choice.
>
> * **Supplementary Investigation**：
> Despite the above reasons, we agree that extending the range of tasks and algorithms is reasonable.
> Therefore, we have conducted a supplementary investigation, which is designed to include two multi-agent competitive tasks in the MPE environment (**Predator-Prey** and **WorldCom**), two classical DRL algorithms (**DDPG** and **MADDPG**), four backdoor tasks, UNIDOOR as the attack method, and three intervention settings (*None*, *Layer Normalization*, and *SAM*).
> The following presents the overall results in post-training scenarios (i.e., *TM-Post*).
> More details on the design, results, and analysis can be found in **Appendix J** of the revised manuscript.
>
> Intervention        | Predator-Prey (ASR ↑) | Predator-Prey (BTP ↑) | WorldCom (ASR ↑) | WorldCom (BTP ↑) |
> |------------------|-------------------|-------------------|----------------|----------------|
> |*None*             | 0.202             | 1.000             | 0.255          | 0.962          |
> |*Layer Normalization* | 0.209           | 0.887             | 0.242          | 0.923          |
> |*SAM*              | 0.319             | 0.993             | 0.409          | 0.960          |

---

### Author Response · Authors · 2025-11-25
**Response Overview**

Hello Area Chair,

We sincerely appreciate the dedication shown by you and all the Area Chairs in response to the recent unforeseen event.
For your convenience in reviewing this paper, we prepare a brief summary of our previous rebuttal discussions with the reviewers:

* **Rebuttal Progress.**

  (1) On 26 Nov 2025, 00:56 (AOE), we received feedback from Reviewer Y4ax indicating that our responses had addressed their concerns and that the revised manuscript improved the quality as requested. Reviewer Y4ax stated: "Overall I would support this paper for publication as a poster unless other reviewers convince me otherwise", and updated their ratings as follows: Rating: 4 → 6, Soundness: 2 → 3, Presentation: 2 → 3.

  (2) While we have not yet received responses from Reviewers bCoi Gy5W and q1rd, we remain grateful for the time and valuable comments provided by all reviewers.
  We have provided detailed responses to all reviewer concerns and firmly believe that their feedback has been invaluable in improving the manuscript, enhancing its clarity, depth, and completeness.
  We also appreciate some reviewers' recognition of our work's motivation, the distilled insights, and the extensive evaluation scale.

* **Main Improvements.**

  (1) Presentation:
  * We distilled the results of our comprehensive investigation into five key findings and highlighted them (see **Section 1** and **Section 5**).
  * We strengthened the causal analysis, such as the intrinsic mechanism of *SAM*'s impact and its theoretical justification (see **Section 5.2** and **Appendix I**).
  * We discussed the logical connections among the five findings and their implications for future research (see **Section 5.3**).
  * We restructured figures and provided additional details on the study settings (see **Appendixes B, E, F, G**), making the manuscript clearer, more organized, and easier to read.

  (2) Evaluation:
  *  We incorporated an additional DRL backdoor attack, **SleeperNets**, which expands the total number of tested cases from 10,998 to 14,664.
  *  We conducted a supplementary investigation, covering two tasks in MPE and two additional DRL algorithms (see **Appendix J**).

In the revised manuscript, we have highlighted the main modifications in **blue text** for easier identification.
While we are aware of the considerable workload you face, we would sincerely appreciate it if you could take a moment to review our manuscript, which we believe offers a novel perspective and provides valuable insights into DRL backdoor research.

Finally, we would like to once again express our sincere gratitude to the Area Chairs and all reviewers.

Sincerely,

Authors of Submission #4742

---

### Meta-Review · Area_Chair_ScYa · 2026-01-07

**Summary:**

Reviewers raised substantive concerns about novelty, positioning, and the strength of the conclusions. Reviewer bCoi gives a clear rejection, arguing that several core findings are well known in the backdoor literature, insufficiently contextualized, and over-claimed given limited attack and algorithm coverage. They also questioned whether the experimental setup supports general claims about deep RL. Reviewers q1rd and Y4ax rated the paper slightly below the threshold, noting that while the empirical effort is substantial, many conclusions remain fragmented, weakly validated, or closer to application-level observations than general principles. Reviewer Gy5W rated the paper slightly above threshold with mainly presentation concerns.

Overall, despite addressing an important problem, the current form falls short of the acceptance standard due to concerns about novelty and mechanistic claims remain relatively unresolved.

**Reviewer Concerns:**

Concerns addressed:
- Presentation and clarity: Multiple reviewers’ complaints about figure readability, confusing acronyms, and fragmented structure were clearly addressed. The revised manuscript is easier to follow, with clearer terminology and highlighted findings.
- Attack coverage: The addition of SleeperNets and a supplementary investigation modestly broadened the evaluation.
- Clarification of metrics and confusing results: These issues were satisfactorily explained.

Concerns that remain outstanding:
- Novelty and positioning: The rebuttal does not convincingly address reviewers’ concerns that many reported results are already known or expected in the broader backdoor-defense literature, leaving the overall contribution limited.

**Reviewer Scores:**

- Reviewer bCoi: Likely remains unchanged at  2. The rebuttal does not fully address their core novelty and positioning issues.
- Reviewer Gy5W: Likely remains unchanged at 6.
- Reviewer q1rd: Likely remains unchanged at 4.
- Reviewer Y4ax: Update from 4 to 6 after rebuttal and would likely remain at 6.

---

### Decision · Program_Chairs · 2026-01-26

Reject